# Shedding Light on Time Series Classification using Interpretability Gated Networks

**Yunshi Wen**[1]*, **Tengfei Ma**[2]*, **Ronny Luss**[3], **Debarun Bhattacharjya**[3],
**Achille Fokoue**[3], **Agung Julius**[1]

[1]Rensselaer Polytechnic Institute, [2]Stony Brook University, [3]IBM Research
`weny2@rpi.edu, tengfei.ma@stonybrook.edu`
`{rluss, debarunb, achille}@us.ibm.com`
`agung@ecse.rpi.edu`

## Abstract

In time-series classification, interpretable models can bring additional insights but be outperformed by deep models since human-understandable features have limited expressivity and flexibility. In this work, we present InterpGN, a framework that integrates an interpretable model and a deep neural network. Within this framework, we introduce a novel gating function design based on the confidence of the interpretable expert, preserving interpretability for samples where interpretable features are significant while also identifying samples that require additional expertise. For the interpretable expert, we incorporate shapelets to effectively model shape-level features for time-series data. We introduce a variant of Shapelet Transforms to build logical predicates using shapelets. Our proposed model achieves comparable performance with state-of-the-art deep learning models while additionally providing interpretable classifiers for various benchmark datasets. We further show that our models improve on quantitative shapelet quality and interpretability metrics over existing shapelet-learning formulations. Finally, we show that our models can integrate additional advanced architectures and be applied to real-world tasks beyond standard benchmarks such as the MIMIC-III and time series extrinsic regression datasets.

## 1 Introduction

Time series classification is a fundamental task for time series (TS) data. Depending on the number of observed variables, TS classification problems can be categorized as either univariate or multivariate. Both categories are involved in a wide range of applications. Univariate TS data includes image outlines, sound, and spectrographs, and multivariate TS data include electroencephalogram (EEG), electrocardiogram (ECG), and human activity recognition (HAR). These types of TS data appear in essential applications such as healthcare, neuroscience, and automation. In recent years, there has been an increasing trend of employing deep learning models to TS classification problems. However, despite achieving state-of-the-art performance, deep models typically lack interpretability.

As a motivating example for interpretable TS classification, consider the problem of classifying ECG data using machine learning (Neves et al., 2021; Abdullah et al., 2023; Aziz et al., 2021). While models might be accurate, doctors may hesitate to rely on them without understanding why a patient's heartbeat as measured by an ECG was classified as being indicative of reduced bloodflow to the heart (known as myocardial ischemia). Our model seeks to provide such trust by discovering classifiers based on understandable concepts, such as those illustrated in Figure 1 (middle), where heartbeats are classified according to the existence of patterns known as shapelets. In this example, shapelet $s_{11}$, representing a downward trend, and $s_{12}$, representing an upward trend, serve as discriminative patterns that classify normal and ischemic heartbeats. A doctor may gain confidence in seeing such patterns with straightforward interpretations, such as in Figure 1 (right), that were deemed important to the predictions.

---

*Most of the work was done while at IBM Research. Corresponding to Yunshi Wen (`weny2@rpi.edu`) and Tengfei Ma (`tengfei.ma@stonybrook.edu`).

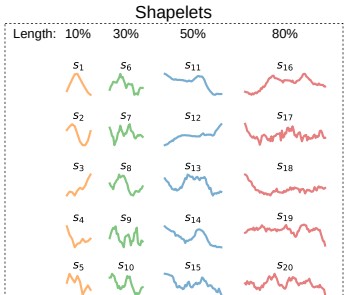 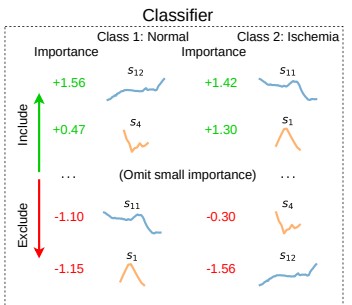 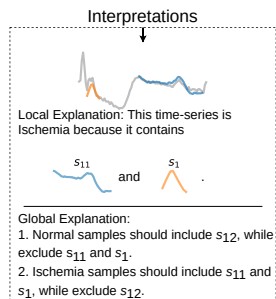

Figure 1: An example of shapelet-based interpretable TS classification on the *ECG200* dataset. (Left) The set of shapelets with various lengths to capture features with different scales. (Middle) A classifier that associates the shapelets with categories, assigning importance scores to them. (Right) Two methods to interpret the classifier result in rule-like explanations.

Interpretability can be significant from two perspectives. Firstly, in applications such as healthcare and physical systems where safety is essential, interpreting a model allows us to validate reliability, trustworthiness, and fairness. Secondly, in challenging tasks, we can gain knowledge about the problem by interpreting patterns found by machine learning models during training. Existing interpretable time series methods (Zhao et al., 2023) include *post hoc* explainability which explain the behavior of a trained deep learning model often using a surrogate model (Queen et al., 2023), and self-explainable models for which model predictions are directly interpretable.

In this study, we focus on self-explainable models. A recent self-explainable TS model, Neuro-Symbolic Time Series Classification (NSTSC) (Yan et al., 2022), combines a neuro-symbolic model (Riegel et al., 2020) with signal temporal logic (Mehdipour et al., 2021); soft-logic predicates are defined on each time stamp, and logic rules are learned to describe the classes. Although such approaches are defined to be interpretable, identifying the logical rules when applied to complex tasks can result in numerous predicates and logical operators. Furthermore, it is known that interpretability may lead to lower classification performance due to incomplete coverage of interpretable concepts or limited expressivity (Koh et al., 2020; Oikarinen et al., 2022; Havasi et al., 2022); such is the case for Concept Bottleneck Models (CBMs) (Koh et al., 2020), a popular interpretable model in computer vision. Similarly in TS, time-domain features (such as shapes and trends) are easy to understand but may not be able to capture other essential features such as those in the frequency domain.

In addition to post-hoc explanations, the literature on combining black-box models with interpretable models includes approaches such as performing logical reasoning on concepts discovered by deep models (Lee et al., 2022) or using deep models to select from a set of interpretable primitives, which are then composed to form the final predictions (Okajima & Sadamasa, 2019). In this paper, with the goal of maintaining interpretability when appropriate while allowing for additional complexity where concept-level information is not significant, we propose an Interpretability Gated Network (InterpGN) that combines an interpretable expert with a deep neural network (DNN) expert. InterpGN uses a novel gating function to decide whether to use the interpretable expert's prediction or to rely on the DNN's expertise (as likely needed on challenging samples). Our interpretable expert, termed the Shapelet Bottleneck Model (SBM), introduces a variant of the Shapelet Transform (Lines et al., 2012) to build expressive yet interpretable predicates based on learned shapelets (Ye & Keogh, 2011) such as the patterns in Figure 1 (left). Our interpretable expert holds several advantages over existing methods such as fewer parameters than neuro-symbolic method NSTSC, improved shapelet selection due to our Shapelet Transform, and better accuracy due to gating on challenging samples.

Our main **contributions** are summarized as follows: (1) a novel gating function that assigns samples to experts based on the confidence level of the interpretable expert, (2) a novel variant of the Shapelet Transform that improves interpretability compared to existing shapelet-based methods, and (3) quantitative metrics for interpretability and shapelet quality, where we show that our models improve in both metrics over existing shapelet-learning formulations. We further demonstrate that InterpGN outperforms state-of-the-art methods on the UEA multivariate TS classification archive (Bagnall et al., 2018), illustrate the interpretability of InterpGN on multivariate TS classification datasets, and finally apply our framework on a real-world healthcare dataset, MIMIC-III (Johnson et al., 2016).

## 2 RELATED WORK

**Machine Learning Methods for TS Analysis** Classical deep learning models such as Multilayer Perceptron (MLP), Fully Convolutional Network (FCN), and Residual Network (ResNet) (Wang et al., 2017) are usually considered as baselines. ROCKET (Dempster et al., 2020) uses random convolution kernels with various lengths to transform TS data and trains a linear classifier. Another direction is representation learning that uses unsupervised approaches to learn from positive and negative samples (Franceschi et al., 2019). For instance, TapNet (Zhang et al., 2020) proposes an attention prototype network to learn multivariate TS representations and address the issue of limited labeled data. Time Series Transformer (TST) (Zerveas et al., 2021) reconstructs TS from masked data to learn a pre-trained model that can be used for multiple types of downstream tasks. Recent works also find that when combined with deep neural networks such as Transformers (Vaswani et al., 2017), the patch-based modeling, which segments TS into channel-independent patches, is more effective than timestamp-based modeling in various tasks (Zuo et al., 2023; Nie et al., 2023; Wu et al., 2023). Although the above methods may achieve state-of-the-art performance, they lack interpretability.

**Interpretable Methods** Pattern-based methods usually have some level of interpretability. (Zhao et al., 2023) surveys many methods for interpretable TS classification. A subset of pattern-based methods use shapelets, defined as discriminatory subsequences of TS data (Ye & Keogh, 2011) selected based on information gain. ShapeNet (Li et al., 2021) learns embeddings that encode subsequences with different lengths into a unified space using triplet loss, and then selects the most representative shapelets to use with Shapelet Transforms. Shapelets can also be learned using gradient-based methods (Grabocka et al., 2014), relaxing the definition of shapelets as sequences with certain patterns. However, with the original formulation, the learned shapelets have poor quality as they may not look similar to the actual subsequences. ADSN (Ma et al., 2020) improves shapelet quality using adversarial training. ShapeConv (Qu et al., 2024) initializes the shapelets as the cluster means of TS subsequence and uses an additional regularization to keep shapelets close to the actual subsequence during training. Besides shapelet-based methods, a recent pattern-based method RLPAM (Gao et al., 2022) first transform TS to a sequence of patterns and uses reinforcement learning to select the most informative patterns for classification.

## 3 PROBLEM FORMULATIONS

**Notation** Let $\mathcal{D} = \{(x_i, y_i)|i = 1, \ldots, N\}$ denotes a TS classification dataset with $N$ samples, where $x_i \in \mathbb{R}^{M \times T}$ is a TS sample and $y_i \in \{1, \ldots, C\}$ is the class label. Here, $M$ is the number of variables, $T$ is the length in timestamps, and $C$ is the number of categories. A multivariate TS sample $x_i$ can be viewed as a set of univariate TS samples where $x_i^m \in \mathbb{R}^T$ denotes the TS of the $m^{\text{th}}$ variable. $x_{i,t_1:t_2}^m$ denotes a subsequence of $x_i^m$ between timestamps $t_1$ and $t_2$.

**Shapelets** Shapelets were originally defined as discriminative TS subsequences to distinguish different categories (Ye & Keogh, 2011). Learning Time-series Shapelets (LTS) (Grabocka et al., 2014) combines Shapelet Transform with gradient-based optimization to learn a set of shapelets from scratch, where the definition of shapelets is relaxed to be short sequences with certain patterns. In this work, we follow the definition of LTS where the shapelets are the model's parameter to train.

**Shapelet-based TS Modeling** Independently modeling each channel of a multivariate TS has been demonstrated to be an effective approach (Wu et al., 2023; Nie et al., 2023). In this paper, we consider a similar approach by formulating univariate shapelets for each variable separately. Let

$$\mathcal{S} = \{s_k^{m,l}|k = 1, \ldots, K; m = 1, \ldots, M; l \in L\}$$

denote a set of shapelets, where $s_k^{m,l} \in \mathbb{R}^l$ denotes a shapelet on the $m^{\text{th}}$ channel with length $l$ and index $k$, i.e., we learn $K$ shapelets for each possible length and channel. In this paper, we consider shapelets with multiple lengths where $L = \{\max(\lceil \delta T \rceil, 3)|\delta = 0.05, 0.1, 0.2, 0.3, 0.5, 0.8\}$. Therefore, $\mathcal{S}$ has a total of $MK|L|$ shapelets.

**Interpretability** Existing methods gain different levels of interpretability by inputting interpretable features (Zuo et al., 2023) into a simple model such as a linear layer (Ma et al., 2020; Qu et al., 2024)

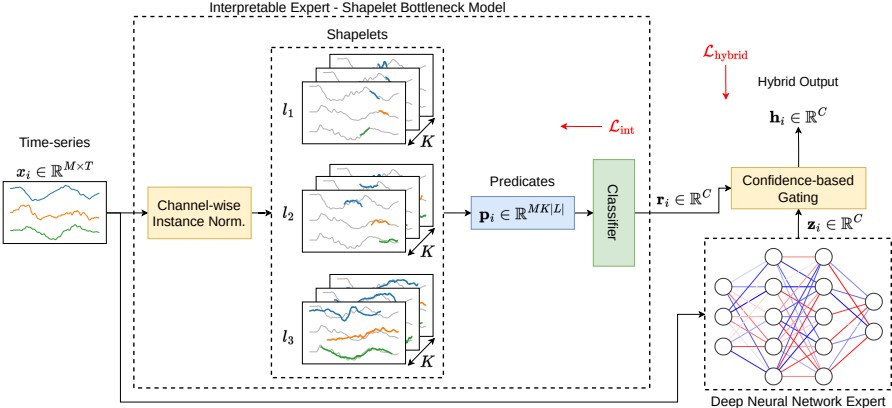

Figure 2: Overview of InterpGN.

or SVM (Li et al., 2021). However, such approaches usually fail to provide explanations of their predictions based on distances features. For the interpretable expert, we build logical predicates using shapelets and the classifier directly provides rule-like explanations. Combining interpretable features with logic allows this approach to select essential shapelets since model weights associated with shapelets represent corresponding importance to predictions (Riegel et al., 2020).

# 4 THE MIXTURE-OF-EXPERTS FRAMEWORK: INTERPGN

In this section, we present the details of Interpretability Gated Networks (InterpGN). We first discuss the proposed variant of Shapelet Transforms to build logical predicates and the formulations of the Shapelet Bottleneck Model (SBM), and then discuss how explanations can be derived from SBM. Lastly, we present the novel gating function used by InterpGN along with the training procedure. An overview of InterpGN is given in Figure 2.

## 4.1 BUILDING LOGICAL PREDICATES USING SHAPELETS

A straightforward way to express predictions with logical classifiers is to use predicates built on features such as distances. Mueen et al. (2011) constructed a logical predicate with a threshold distance; however, when combined with LTS, we find this approach reduces shapelet quality (essential shapelets are not similar to the TS) as well as interpretability (notion of distance can vary across datasets) since the threshold distance does not directly reflect the existence of a shapelet (see Figure 3 for an example). Therefore, analogous to concepts in CBM, we view shapelets as interpretable concepts for TS and build logical predicates directly from shapelet distances. Consider the distance between $s_k^{m,l}$ and $x_i^m$ defined as:

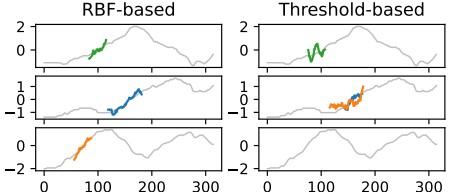

Figure 3: Comparison between shapelets learned by (left) radial basis function (RBF) based predicates and (right) threshold based predicates. The three most important shapelets are visualized.

$$d_{i,k}^{m,l} = \min_{t=1,\dots,T-l} \texttt{dist}(s_k^{m,l}, x_{i,t:t+l}^m), \tag{1}$$

where $\texttt{dist}(s_k^{m,l}, x_{i,t:t+l}^m)$ is the Euclidean distance between the two sequences. We introduce a variant of the Shapelet Transform, formally defined by

$$p_{i,k}^{m,l} = \max_{t=1\dots,T-l} e^{-\left(\epsilon \cdot \texttt{dist}(s_k^{m,l}, x_{i,t:t+l}^m)\right)^2} \tag{2}$$

using a Gaussian radial basis function on the distance to measure the likelihood that $s_k^{m,l}$ exists in $x_i^m$, where $\epsilon$ is a scaling parameter that controls the steepness of the kernel. We say that the

set of shapelets $\mathcal{S}$ transforms each sample $x_i \in \mathcal{D}$ to predicates $\mathbf{p}_i \in \mathbb{R}^{MK|L|}$, where $\mathbf{p}_i = \left[ p_{i,k}^{m,l} \mid k = 1, \dots, K;\, m = 1, \dots, M;\, l \in L \right]$.

## 4.2 SHAPELET BOTTLENECK MODEL - THE INTERPRETABLE EXPERT

When constructing the SBM, we adopt a similar approach to LTS and CBM to preserve interpretability and produce rule-like classifiers. Formally, the predicates $\mathbf{p}_i$ are fed into a linear layer to compute the outputs

$$r_{i,c} = \sum_{k=1,\dots,K;\, m=1,\dots,M;\, l \in L} w_{c,k}^{m,l}\, p_{i,k}^{m,l}, \ \forall c \in \{1, \dots, C\}. \tag{3}$$

The SBM's output is $\mathbf{r}_i \in \mathbb{R}^C = [r_{i,1}, \dots, r_{i,C}]$. An overview of SBM is given in Figure 2. Note that the different shapelet lengths are illustrated vertically under the Shapelet Bottleneck; each row allows shapelets of length $l$ across all variables.

**Training** SBM is end-to-end differentiable and trained using gradient-based optimization. The parameters to be trained include the linear classifier weights $w_{c,k}^{m,l}$ and the shapelets $s_k^{m,l}$, both of which are randomly initialized. In practice, gradients of the `max` operator in Equation 2 can be substituted by gradients of `softmax` using straight-through estimation. The objective function contains the following three loss functions:

- Classification loss: softmax cross-entropy loss $\mathcal{L}_{\text{ce}}$ on $(\mathbf{r_i}, y_i)$,
- Shapelet diversity loss (Ma et al., 2020) to regulate learning redundant shapelets:

$$\mathcal{L}_{\text{div}} = \frac{1}{2KM|L|} \sum_{m=1}^{M} \sum_{l \in L} \sum_{k_1=1}^{K} \sum_{\substack{k_2=1 \\ k_2 \neq k_1}}^{K} e^{-\|s_{k_1}^{m,l} - s_{k_2}^{m,l}\|_2}, \tag{4}$$

- L1 regularization $\mathcal{L}_{\text{reg}}$ on the classifier weights $w_{c,k}^{m,l}$ to encourage sparsity, select the most informative concepts, and produce simple classifiers.

The overall loss function is $\mathcal{L}_{\text{int}} = \mathcal{L}_{\text{ce}} + \lambda_{\text{div}}\mathcal{L}_{\text{div}} + \lambda_{\text{reg}}\mathcal{L}_{\text{reg}}$, where $\lambda_{\text{div}}$, and $\lambda_{\text{reg}}$ are hyperparameters.

## 4.3 INTERPRETATIONS

A concept probability $p_{i,k}^{m,l}$ is a logical predicate whose value measures the existence of shapelet $s_k^{m,l}$. The linear classifier behaves like weighted linear logic analogous to the operators in Riegel et al. (2020), where weights $w_{c,k}^{m,l}$ specify the importance of shapelet $s_k^{m,l}$ to class c. We propose two ways to interpret the model.

**Local Explainability** Local explanations answer why the classification decision for a sample $x_i$ was made. In SBM, the explanation has the form of "sample $x_i$ belongs to class $c$ because $x_i$ contains shapelet $s_k^{m,l}$", whose corresponding $w_{c,k}^{m,l} p_{i,k}^{m,l}$ is significant. Consider Figure 1 in the Introduction where we visualize the two shapelets with the largest $w_{c,k}^{m,l} p_{i,k}^{m,l}$ for each class. Such interpretations allow us to visually validate the correctness of learned shapelets and classifiers. Note that, to the best of our knowledge, most existing shapelet-based methods (Qu et al., 2024) and post-hoc explanation methods (Queen et al., 2023) only provide local explainability.

**Global Explainability** Global explanations provide knowledge about the classification problem, expressed in SBM by inductive logics without pertaining to particular samples. SBM offers the following global explanation: (1) $w_{c,k}^{m,\delta} > 0$: samples in class $c$ *should* contain $s_k^{m,\delta}$, (2) $w_{c,k}^{m,\delta} = 0$: $s_k^{m,\delta}$ is unrelated to class $c$, (3) $w_{c,k}^{m,\delta} < 0$: samples in class $c$ *should not* contain $s_k^{m,\delta}$.

Most figures in this paper visualize shapelets with the highest weights in the linear classifier, validating the correctness of the global explanation. For some challenging tasks, we use local explanations to provide visually better explanations for specific samples. Note the difference between local and global explainability: local explanations discuss what *does* occur in a sample whereas global explanations discuss what *should* occur in a sample.

### 4.4 MIXTURE-OF-EXPERTS GATING

In the previous section, the shapelet bottleneck transforms TS into low-dimensional representations in the form of logical predicates. While the `max` operator in Equation (2) makes the classifier more interpretable, it also reduces expressiveness. For example, the predicates can only measure the existence of shapelets but lose information about the number of occurrences and corresponding timestamps. In TS with discriminative features in other domains, such as the frequency domain instead of the time domain, the shapelet bottleneck may not capture essential features for classification. To address this limitation, we employ a Mixture-of-Experts approach to build a partially-interpretable hybrid model.

A Mixture-of-Experts (MoE) combines the output of multiple expert models with a gating network (Shazeer et al., 2017). Experts have different structures to capture different aspects of the data and the gating network assigns the work to the experts. IME (Ismail et al., 2023) views a simple model (such as linear regression or soft decision tree) as an interpretable gating network. From a different perspective, the interpretable expert itself can serve as the gating network, which reduce the number of parameters and improves interpretability. We introduce a gating function to assign the work to experts based on the confidence level of the interpretable expert.

The outputs $\mathbf{r}_i$ minimizes $\mathcal{L}_{\text{ce}}$ during training. Denote $\hat{\mathbf{r}}_i = \texttt{softmax}(\mathbf{r}_i)$ with components $\hat{r}_{i,c}$ corresponding to Equation (3). Then the optimal $\hat{\mathbf{r}}_i$ is a one-hot vector with $\hat{r}_{i,c} = 1$ if $y_i = c$. Diversity of $\hat{r}_{i,c}$ thus measures confidence of the model. When combining an interpretable model and a DNN with MoE, the use of DNN should be inverse proportional to the confidence level of the interpretable model to maximize interpretability. Therefore, we design the gating function

$$\eta(x_i) = \frac{C \cdot \sum_{c=1}^{C} (\hat{r}_{i,c})^2 - 1}{C - 1}, \tag{5}$$

which is a modified Gini Index that measures the diversity of variables in $\hat{\mathbf{r}}_i$. Intuitively, $\eta(x_i) = 0$ when values in $\hat{\mathbf{r}}_i$ are identical, and $\eta(x_i) = 1$ when $\hat{\mathbf{r}}_i$ is a one-hot vector. During training, the output of the hybrid model is then a mixture of the outputs of the interpretable expert and the DNN with ratio $\eta(x_i)$:

$$\mathbf{h}_i = \mathbf{r}_i \cdot \eta(x_i) + \mathbf{z}_i \cdot (1 - \eta(x_i)), \tag{6}$$

where $\mathbf{z}_i$ is the output of the DNN with input $x_i$. During inference, we activate only the SBM and discard the DNN if the confidence of SBM is high. Specifically, given a gating value $\underline{\eta}$, we set $\eta(x_i) = 1$ if $\eta(x_1) > \underline{\eta}$.

### 4.5 TRAINING OF INTERPGN

The overall objective of the InterpGN framework is to minimize $\mathcal{L}_{\text{hybrid}} = \beta \mathcal{L}_{\text{int}} + \bar{\mathcal{L}}_{\text{ce}}$. Here, $\bar{\mathcal{L}}_{\text{ce}}$ is another softmax cross-entropy loss on $(\mathbf{h}_i, y_i)$ to optimize the overall performance of the hybrid model. $\beta \in [0, 1]$ is a scalar to weight the training of the interpretable expert, which is set to be either a constant hyperparameter or decaying with a cosine schedule. Intuitively, the $\mathcal{L}_{\text{int}}$ term directly optimizes the SBM component, which ensure that the SBM is actively encouraged to find meaningful shapelets. This design prevents InterpGN from collapsing into a pure DNN model whenever the shapelet-based features are useful (see Figure 18 and Figure 19). The hybrid model is still end-to-end differentiable and trained using gradient-based optimizations methods.

## 5 EXPERIMENTS

We first compare InterpGN against various methods on the UEA multivariate TS classification archive, presenting insights gained from visualizing concept representations. Results from SBM without the additional expert are also given. We then demonstrate the benefits of our models on the real-world application of in-hospital mortality early prediction (Harutyunyan et al., 2019) using the MIMIC III dataset. We conclude with ablation studies on the effects of each component in our model. As a basic InterpGN design, we combine the SBM with a Fully Convolutional Network (FCN) since it has best performance among the three basic deep models for TS classification (Wang et al., 2017). Details of the experimental setup are provided in Appendix A. [1]

---

[1] Code is available at: `https://github.com/YunshiWen/InterpretGatedNetwork`.

Table 1: Comparison on 30 datasets from the UEA archive. The best accuracy for each dataset is marked with bold, the second-best is marked with italic, and the third-best is marked with underline.

| | Classical | | Deep Learning | | | | | | With Interpretable Features | | | Ours | |
|---|---|---|---|---|---|---|---|---|---|---|---|---|---|
| Dataset | STRF | DTW | FCN | TS2Vec | TimesNet | TST | PatchTST | SVP-T | ShapeNet | RLPAM | ShapeConv | SBM | InterpGN |
| Avg. Accuracy | 0.645 | 0.650 | *0.746* | 0.704 | 0.702 | 0.709 | 0.611 | 0.730 | 0.697 | 0.740 | 0.743 | 0.726 | **0.760** |
| Avg. Rank | 9.700 | 9.138 | *5.000* | 7.533 | 7.517 | 6.778 | 10.414 | 5.267 | 6.567 | 5.400 | 5.500 | 5.733 | **3.500** |
| Num. Top-1 | 2 | 1 | 4 | 0 | 1 | 4 | 0 | 5 | 2 | 7 | 2 | 3 | **8** |
| Num. Top-3 | 2 | 1 | *14* | 2 | 3 | 9 | 1 | 11 | 5 | 13 | 10 | 8 | **16** |
| Wins/Draws | 28 | 27 | 23 | 27 | 25 | 18 | 27 | 19 | 25 | 20 | 20 | 20 | - |
| Losses | 2 | 2 | 7 | 3 | 4 | 9 | 2 | 11 | 5 | 10 | 10 | 10 | - |
| Wilcoxon p-value | 0.000 | 0.000 | 0.003 | 0.000 | 0.000 | 0.001 | 0.000 | 0.066 | 0.001 | 0.112 | 0.009 | 0.003 | - |

## 5.1 MULTIVARIATE TIME SERIES CLASSIFICATION

First, we evaluate the performance of InterpGN on 30 datasets from the UEA multivariate TS classification archive and compare with baseline methods including (1) classical search-based method: Shapelet Transform with Random Forest Classifier (STRF) and DTW (Chen et al., 2013), (2) black-box deep learning methods: FCN (Wang et al., 2017), TS2Vec (Yue et al., 2022), TimesNet (Wu et al., 2023), TST (Zerveas et al., 2021), PatchTST (Nie et al., 2023), SVP-T (Zuo et al., 2023), and (3) methods with interpretable features: ShapeNet (Li et al., 2021), RLPAM (Gao et al., 2022), ShapeConv (Qu et al., 2024).

Table 1 presents average classification accuracy across the UEA datasets; the full results are included in Table 5 of Appendix B.1. Based on the average rank across the 30 datasets, InterpGN outperforms the baseline methods and SBM achieves comparable performance without help from the FCN.

### 5.1.1 INTERPRETABILITY

In addition to its effective performance, SBM provides interpretable logical classifiers. Figure 4 visualizes shapelets learned by SBM on a multivariate TS dataset from global explanations, where the blue shapelets have the highest positive weights and the red shapelets have the lowest negative weights. Figure 4 visually confirms that the models capture discriminative shapelet features.

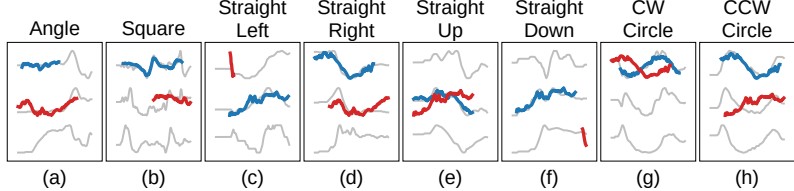

Figure 4: Visualization of the *UGL* dataset from the UEA archive with accelerometer readings of eight gestures. For example, in $(g)$, the classifier can be interpreted as: a "CW Circle" gesture should contain the blue shapelet but not the red shapelet in channel 1.

### 5.1.2 SHAPELET BOTTLENECK REPRESENTATION

We explore the effectiveness of the shapelet predicates by viewing probabilities $\mathbf{p}_i$ as representations for TS data. Figure 5 displays the EP dataset using the dimensionality reduction technique t-SNE (van der Maaten & Hinton, 2008). Representations learned by SBM and InterpGN are clearly more separable than the raw TS data. While a few samples are mixed into incorrect clusters (e.g., blue mixed into green), the DNN in the InterpGN model can make corrections and result in better accuracy than SBM. Shading intensity of data points in Figure 5 (c) represents $\eta(x_i)$. Intuitively, predictions of shaded points are interpretable as they are based on SBM while lighter points rely more on the DNN. We observe that most transparent points lie along the cluster boundaries, meaning InterpGN learns to use interpretable logical classifier for easier-to-classify samples while relying on the DNN to classify the difficult boundary samples which can be identified as requiring additional expertise. In Figure 17, we further present the behavior of InterpGN by visualizing the distribution of $\eta(x_i)$. We can conclude that the predictions of InterpGN is consistent with SBM when $\eta(x_i)$ is high, preserving interpretability for those samples.

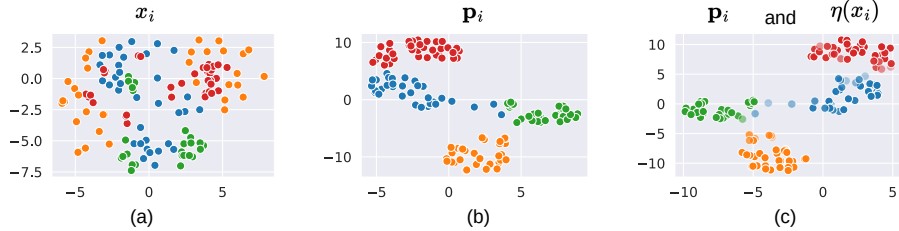

Figure 5: t-SNE dimensionality reduction of (a) raw data $x_i$, (b) $\mathbf{p}_i$ of SBM, and (c) $\mathbf{p}_i$ of InterpGN on the *EP* dataset. Each color represents samples with a specific label. Samples with low $\eta(x_i)$ from InterpGN lie on the boundary, i.e., are more difficult to predict and can benefit from additional expertise.

## 5.2 PREDICTING IN-HOSPITAL MORTALITY

The MIMIC-III dataset contains electronic healthcare records from ICU patients. We follow the procedure in (Harutyunyan et al., 2019) to preprocess and generate data to predict in-hospital mortality based on patient records from the first 48 hours after entering an ICU. Setup details are provided in Appendix A.2. Table 2 shows results illustrating how InterpGN helps SBM on the difficult samples.

**Interpretability** Clinical records from the MIMIC-III dataset are challenging due to sample diversity and missing values. Figure 6 highlights an example of local explanations on two samples. For better representation, we visualize the five most essential shapelets on their corresponding channels and omit other channels. The model captures long-term discriminative trends in three channels, HR, MBP and OS. The shapelets and classifier can be interpreted as: the

Table 2: Results for in-hospital mortality predication on the MIMIC-III dataset.

| Metric | STRF | FCN | SBM | InterpGN |
|---|---|---|---|---|
| Accuracy | 0.653 | 0.693 | 0.658 | **0.703** |
| F1 | 0.639 | **0.675** | 0.657 | 0.657 |
| Recall | 0.615 | 0.634 | **0.657** | 0.569 |
| Precision | 0.666 | 0.734 | 0.659 | **0.784** |
| ROC-AUC | 0.653 | 0.698 | 0.658 | **0.703** |

first patient passed away due to decreasing HR, increasing MBP, and drop in OS; whereas the second survived because of increasing HR after a local drop, rapid decrease in MBP, and steady OS. In real-world scenarios, these interpretations could bring more insights to clinical applications than DNNs that only provide black-box predictions.

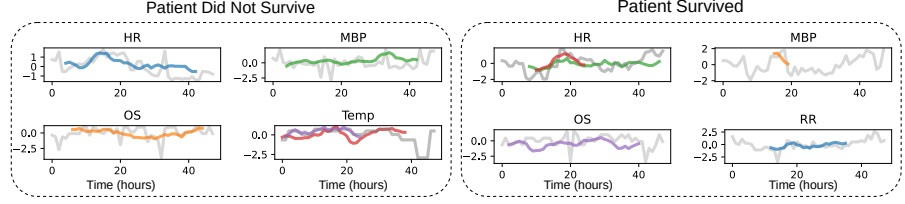

Figure 6: Local explanations of SBM for in-hospital mortality predication. The variables in the figure are: Heart Rate (HR), Mean Blood Pressure (MBP), Oxygen Saturation (OS), Respiratory Rate (RR), and Temperature (Temp).

## 5.3 EFFECTS OF HYPERPARAMETERS ON INTERPRETABILITY AND SHAPELET QUALITY

Analogous to the results presented in Figure 1 and Section 5.1.1, existing interpretable models mainly demonstrate qualitative results on interpretability by visualizing the interpretable features (Ma et al., 2020; Qu et al., 2024). However, there remains a lack of quantitative metrics to measure the level of interpretability and quality of features. Therefore, we introduce two additional metrics to quantitatively assess the interpretable models.

**Interpretability metric**: Naturally, the prediction from the model is considered more interpretable if fewer features are needed to simulate and understand the prediction. With a linear classifier taking logical predicates as inputs, interpretability of SBM can be measured by the sparsity of the classifier weights since fewer shapelets contribute to the prediction as weight sparsity increases. As a metric for sparsity, we choose three empirical thresholds and compute the ratio of weights above each threshold. Formally, the sparsity metrics are defined as

$$\text{ratio} = \frac{1}{|\mathcal{W}|} \sum_{w \in \mathcal{W}} \mathbb{1}(|w| > t), \textbf{ for } t \in \{1, 0.5, 0.1\}. \tag{7}$$

Additionally, as a relative sparsity measurement, we also compute the Gini coefficient of $|w|$.

**Shapelet quality metric**: In order to produce reliable representations in shapelet-based models, the shapelets need to be "close" to the actual TS subsequences. Therefore, shapelet quality can be measured by shapelet distances (defined by Equation 1). In a prediction, more important shapelets should be expected to have higher quality. Therefore, for our SBM, the shapelet quality metric is defined as

$$\text{shapelet error} = \frac{\mathbb{1}(\hat{y}_i = y_i)}{MK|L|} \sum_{m=1}^{M} \sum_{l \in L} \sum_{k=1}^{K} \max(0, w_{y_i,k}^{m,l}) d(x_i^m, s_k^{m,l}), \ (x_i, y_i) \in \mathcal{D}_{\text{test}} \tag{8}$$

which measures the shapelet distances for samples that are correctly classified, averaged over all shapelets in SBM and weighted by their corresponding classifier weight if $w_{c,k}^{m,l} > 0$. Low shapelet error reflects that the essential shapelets for the prediction are close to the actual TS subsequences which represent high shapelet quality.

**Results and Analysis**   We conduct an ablation study on the effect of hyperparameters on classification accuracy, interpretability, shapelet quality, and interpretable expert utility. Details for the ablation study setup are summarized in Appendix A.4. The results are summarized in Tables 6, 7, 8 and Figures 20, 21, 22, 23, which lead to the following respective conclusions:

• **Number of shapelet** $K$ determines the expressiveness of SBM. Table 6 shows that increasing $K$ improving accuracy while additionally improving shapelet quality since the shapelet could capture more sample-specific patterns. Although the classifier weights also become sparser, the absolute number of weights above the threshold values may also increase. Therefore, the effect of $K$ on interpretability may not be clearly concluded. In InterpGN, the SBM utility rate $\eta$ also increase as $K$ increase since the SBM becomes more expressive.

• **Schedule of of loss weighting** $\beta$ controls the training of the interpretable expert in InterpGN. If $\beta$ decays with a cosine schedule, the SBM no longer trains on samples that cannot be confidently predicted as the models get trained. Table 7 shows that, compared to a constant schedule, models trained with the cosine schedule result in slightly worse accuracy but improve in both shapelet quality and interpretability since the SBMs focus more on samples where shapelet features are useful. As less samples are fitted, $\eta$ also decreases.

• **Predicate type** defines the method of building logical predicates from shapelet distances. In our models, the SBM uses the RBF predicates defined in Equation 2. Another commonly used formulation is a linear predicate (Mueen et al., 2011; Yan et al., 2022) which computes threshold-based predicates by $p_{i,k}^{m,l} = \texttt{sigmoid}(c_k^{m,l} - d_{i,k}^{m,l})$ where $c_k^{m,l}$ is the threshold value to learn. Table 8 shows that our RBF predicates outperform linear predicates in accuracy, shapelet quality, and interpretability.

• $\epsilon$ **affects the steepness of the RBF kernels**. A large $\epsilon$ defines a tight kernel and pushes the shapelet to be close to the TS data. Figure 20 demonstrates that increasing $\epsilon$ improves shapelet quality in the SBM but significantly reduces the accuracy for $\epsilon \geq 5$. $\epsilon = 1$ offers the best balance between shapelet quality and accuracy.

• **Weight regularization** $\lambda_{\textbf{reg}}$ affects the sparsity of classifier weights and the number of shapelets contributing to the predictions. Figure 23 shows that increasing $\lambda_{\text{reg}}$ improves interpretability without significantly affecting the accuracy for $0 \leq \lambda_{\text{reg}} \leq 1$.

• **Shapelet diversity regularization** $\lambda_{\textbf{div}}$ encourages learning different shapelets. Figure 22 leads us to conclude that $\lambda_{\text{div}}$ does not have a significant effect on the three metrics.

• **Gating value** $\eta$ controls the threshold of discarding the DNN during inference. Prediction of samples with $\eta(x_i) > \underline{\eta}$ only rely on the SBM, maximizing their interpretability. Figure 21 shows that reducing the gating value results in slight degrade in accuracy, where the difference in average accuracy between using $\underline{\eta} = 0.5$ and $\underline{\eta} = 1$ is 0.0035.

## 5.4 EXTENDED EXPERIMENTS

Finally, we provide preliminary results for more extensive experiments, including the following:

• **Architecture of the DNN expert within the InterpGN**: We experiment with four supervised deep learning models: FCN, Transformer, PatchTST, and TimesNet (see Appendix C.1). Among the tested architectures, InterpGN with FCN achieves the highest accuracy on most datasets. Interestingly, we observe that the choice of DNN significantly affects $\eta$ for each dataset, reflecting the compatibility between the SBM and the DNN. This highlights a complex relationship between the features learned by the SBM and those captured by different DNN architectures, which may be worth more in-depth study.

• **Shapelet distance metric and classifier architecture in the SBM**: For the shapelet distance function `dist`, we further compare cosine similarity and Pearson correlation with the conventional Euclidean distance (see Appendix C.2). For the classifier in SBM, we compare bi-linear and attention-based classifiers, which could model more complex relationships between shapelets, with the conventional linear classifier (see Appendix C.3). These variants sometimes yield better performance on some datasets, but not consistently, indicating that they could be meaningful architecture choices to tune for each dataset.

• **Faithfulness of shapelets learned by SBM**: We follow Alvarez Melis & Jaakkola (2018) to assess the faithfulness of shapelets learned by the SBM, which measures the correlation between the presence of essential shapelets and the predictions of SBM (see Appendix B.2.1). Over the 30 UEA datasets, we observe high faithfulness on most of the datasets. Notably, all datasets have faithfulness greater than 0, indicating that the learned shapelets are always positively correlated with the predictions.

• **Time series extrinsic regression**: To further examine the generalizability and limitations of the proposed methods, we apply them to time series extrinsic regression tasks (Tan et al., 2021). In order to leverage classification models for regression, we train and evaluate them using an ordinal regression framework, employing the Continuous Ranked Probability Score (CRPS) (Gneiting & Raftery, 2007) as the performance metric. Additional details are provided in Appendix C.4.

## 6 CONCLUSION AND LIMITATIONS

In this paper, we address the limitations of existing TS classification methods. We introduce InterpGN, which leverages the trade-off between interpretability and performance. We present a variant of the Shapelet Transform to build logical predicates from shapelets and propose a novel gating function design based on confidence of the interpretable model. Our experiments demonstrate that InterpGN outperforms existing methods while preserving interpretability for most samples. We also interpreted and analyzed patterns and classifiers learned by the models on various datasets, and applied our models to the MIMIC-III dataset to provide interpretable features and discover knowledge on a real-world task. In the future, we plan to extend the shapelet-based approaches to other time series analysis tasks (Ansari et al., 2024) and build scalable interpretable models (Wen et al., 2024).

We note two **limitations** of our study. Firstly, in some classification rules, the most important shapelet for class $c$ may not look similar to subsequences of TS in class $c$. Such rules tend to have interpretations with form "$x_i$ belongs to class $c$ since it does not contain the essential shapelets for $c' \neq c$" or "$x_i$ belongs to class $c$ since other $c' \neq c$ do not contain shapelet $s$". For example, the blue shapelet in Figure 4 (a) may not capture the essential trend for the "Angle" movement but it can still be correctly categorized since it does not contain the essential shapelets for other categories. Secondly, note that we have studied a simple design which includes one interpretable expert and one DNN. Analogous to the MoE (Shazeer et al., 2017) framework, InterpGN can scale up by integrating more experts. For example, multiple interpretable models can be included using a hierarchical gating approach, while multiple DNN models can be combined using the original MoE approach.

ACKNOWLEDGMENT

This work is funded through the IBM-RPI Future of Computing Research Collaboration. We thank the teams at UCR, UEA, and Monash University for their efforts in creating and publicly sharing the benchmark datasets. We also appreciate Wu et al. for providing the open-source time series benchmarking repository. Finally, we are grateful to the anonymous reviewers of NeurIPS 2024 and ICLR 2025 for their insightful and constructive feedback, which greatly enhanced the quality and scope of our work.

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

## A EXPERIMENTAL SETUP

**Environment.** All the experiments are implemented using Python 3.11 and PyTorch 2.4.0. All the experiments run on a NVIDIA V100-SXM2-32GB GPUs. The SBM and InterpGN are trained on a single GPU. The STRF is implemented using Random Shapelet Transforms from the *Aeon Toolkit* [2] and Random Forest Classifiers from the *Scikit-learn* package, which runs on CPU using 64 threads.

### A.1 UEA MULTIVARIATE TIME-SERIES CLASSIFICATION ARCHIVE

The UEA multivariate TS classification archive (Bagnall et al., 2018) contains 30 datasets collected from various domains. Details of the 30 datasets are provided in Table 4. Data preprocessing and training procedure follow the benchmarking repository provided by Wu et al. (2023) [3]. The default hyperparameter to produce results in Table 1 and Table 5 are summarized in Table3. For each dataset, we train the models on the default training split with 5 different random seeds and report the average accuracy on the test split. Statistics on experiment variations could refer to Appendix B.4.

Table 3: Default hyperparameters.

| Hyperprameter | Default value |
| --- | --- |
| Number of shapelet | $K = 10$ |
| Steepness of RBF kernels | $\epsilon = 1$ |
| Diversity loss weight | $\lambda_{\text{div}} = 0.1$ |
| Sparsity loss weight | $\lambda_{\text{reg}} = 0.1$ |
| $\beta$ schedule | constant |
| Inference gating value | $\eta = 1$ |
| Optimizer | Adam |
| Learning rate | 0.005 |
| Weight decay | 0 |
| Batch size | 32 |

The metrics in Table 1 and Table 5 are:

- **Avg. Accuracy**: Average test accuracy.
- **Avg. Rank**: Average ranking of the method among all baseline methods.
- **Num. Top-1**: Number of datasets the method performs the best among all baseline methods.
- **Num. Top-3**: Number of datasets the method achieves top-3 accuracy among all baseline datasets.
- **Wins/Draws**: Number of datasets that InterpGN achieves better/same accuracy compared to other baseline methods.
- **Looses**: Number of datasets that InterpGN achieves worse accuracy compared to other baseline methods.
- **Wilcoxon p-value**: p-value in the Wilcoxon signed-rank test, which measures if the performance of InterpGN is statistically significant compared to the baseline methods. A small p-value indicates significance, while a large p-value indicates the InterpGN and the baseline method have comparable performance on the UEA datasets.

---

[2]The Aeon Toolkit is available at `www.aeon-toolkit.org`
[3]The benchmarking repository is available at `github.com/thuml/Time-Series-Library`

Table 4: Information of the 30 benchmark datasets from the UEA archive.

| Dataset | Dataset Code | Train Size $N_{\text{train}}$ | Test Size $N_{\text{test}}$ | Variables $M$ | Length $T$ | Categories $C$ |
|---|---|---|---|---|---|---|
| ArticularyWordRecognition | AWR | 275 | 300 | 9 | 144 | 25 |
| AtrialFibrillation | AF | 15 | 15 | 2 | 640 | 3 |
| BasicMotions | BM | 40 | 40 | 6 | 100 | 4 |
| CharacterTrajectories | CT | 1422 | 1436 | 3 | 119 | 20 |
| Cricket | CK | 108 | 72 | 6 | 1197 | 12 |
| DuckDuckGeese | DDG | 50 | 50 | 1345 | 270 | 5 |
| ERing | ER | 30 | 270 | 4 | 65 | 6 |
| EigenWorms | EW | 128 | 131 | 6 | 17948 | 5 |
| Epilepsy | EP | 137 | 138 | 3 | 206 | 4 |
| EthanolConcentration | EC | 261 | 263 | 3 | 1751 | 4 |
| FaceDetection | FD | 5890 | 3524 | 144 | 62 | 2 |
| FingerMovements | FM | 316 | 100 | 28 | 50 | 2 |
| HandMovementDirection | HM | 160 | 74 | 10 | 400 | 4 |
| Handwriting | HW | 150 | 580 | 3 | 52 | 26 |
| Heartbeat | HB | 204 | 205 | 61 | 405 | 2 |
| InsectWingbeat | IW | 25000 | 25000 | 200 | 22 | 10 |
| JapaneseVowels | JV | 270 | 370 | 12 | 29 | 9 |
| LSST | LSST | 2459 | 2466 | 6 | 36 | 14 |
| Libras | LB | 180 | 180 | 2 | 45 | 15 |
| MotorImagery | MI | 278 | 100 | 64 | 3000 | 2 |
| NATOPS | NT | 180 | 180 | 24 | 51 | 6 |
| PEMS-SF | PM | 267 | 173 | 263 | 144 | 7 |
| PenDigits | PD | 7494 | 3498 | 2 | 8 | 10 |
| PhonemeSpectra | PH | 3315 | 3353 | 11 | 217 | 39 |
| RacketSports | RS | 151 | 152 | 6 | 30 | 4 |
| SelfRegulationSCP1 | SCP1 | 268 | 293 | 6 | 896 | 2 |
| SelfRegulationSCP2 | SCP2 | 200 | 180 | 7 | 1152 | 2 |
| SpokenArabicDigits | SAD | 6599 | 2199 | 13 | 93 | 10 |
| StandWalkJump | SWJ | 12 | 15 | 4 | 2500 | 3 |
| UWaveGestureLibrary | UGL | 120 | 320 | 3 | 315 | 8 |

## A.2 PREDICTING IN-HOSPITAL MORTALITY ON MIMIC-III

From the original MIMIC-III dataset, we first extract time-series and impute missing values using the method proposed by Harutyunyan et al. (2019). For predicting in-hospital mortality, the data is highly imbalance, with more than 80% positive samples (patient survived). Therefore we randomly select a subset of 1500 positive and 1500 negative samples to evaluate our models. A sample in the dataset is a TS with $M = 9$ and $T = 48$. We further divide the subset into 80% training and 20% validation to evaluate the performance using 5-fold cross validation. Omitting categorical variables in the MIMIC-III datasets, the TS data has the following 9 variabels:

1. Diastolic blood pressure (DBP),

2. Fraction inspired oxygen (FIO),

3. Heart Rate (HR),

4. Mean blood pressure (MBP),

5. Oxygen saturation (OS),

6. Respiratory rate (RR),

7. Systolic blood pressure (SBP),

8. Temperature (Temp),

9. Glucose (Glu).

### A.3 LIMITATION IN IMPLEMENTATION

When combining shapelet learning with modern deep learning frameworks such as PyTorch, the learning suffers from a memory overhead. Intuitively, the transformations in Equation 1 and Equation 2 slide $s_k^{m,l}$ over $x_i$ and take the minimum/maximum, which is analogous to a convolution layer with pooling. In practice, for efficiency concern, the memory cost of this operation is $O(T \cdot l)$. When $T$ is large (e.g. $T > 2000$) this operation requires numerous GPU memory, mainly due to the unnecessary intermediate variables and gradients are retained . However, the traditional convolution operation does not suffer from this because of low-level optimizations from CUDA, which does not retain the intermediate gradients. Note that ShapeConv (Qu et al., 2024) makes a connection between shapelet transform and convolution by:

$$d(x_i^m, s_k^{m,l}) = -2 \max_{j=1,\dots,T-l+1} \left[ s_k^{m,l} * x_i^m - \frac{\|s_k^{m,l}\|_2^2 + \|x_i^m\|_2^2}{2} \right],$$
(9)

where $s_k^{m,l} * x_i^m$ is the convolution (cross-correlation) operation. However, we found that, in actual implementations, the additional norm term still suffers from the memory overhead.

To address this memory limitation on very long sequences, we provide two implementations for the time-memory trade-off:

1. A time-efficient implementation using large matrix operations:

```
x = x.unfold(2, shapelet_length, 1)
x = x.permute((0, 2, 1, 3)).unsqueeze(2)
d = dist(x - shapelets).mean(dim=-1)
```

2. A memory-efficient implementation using loops:

```
class ShapeletDistanceFunc(torch.autograd.Function):
    def forward(ctx, x, s):
        ctx.save_for_backward(x, s)
        output = torch.cat([dist(s - x[:, :, i:i+s.shape[-1]])
            for i in range(x.shape[-1] - s.shape[-1]+1)], dim=1)
        return output
    def backward(ctx, grad_output):
        # Compute gradient of s
        return 0, grad_s

def ShapeletDistance(x, s):
    return ShapeletDistanceFunc.apply(x, s)

d = ShapeletDistance(x, shapelet)
```

On the largest datasets, this memory-efficient implementation takes less than 12 GB of GPU memory, but requires significantly more time to compute.

Another method to mitigate this issue is by using stride. For example, compared to traditional convolution operation `Conv1d()`, $s_k^{m,l}$ transforms $x_i$ into $p_{i,k}^{m,l}$ using Equation 2 with $\mathtt{in\_channels} = 1$, $\mathtt{out\_channels} = 1$, $\mathtt{kernel\_size} = l$, $\mathtt{stride} = 1$, $\mathtt{padding} = 1$, and $\mathtt{dilation} = 1$. Memory cost reduces as `stride` increase. When $l$ is large, this would not bring significant effects since the learned shapelets are abstracted patterns. In practice, we find that setting $\mathtt{stride} = \log(l)$ results in the same level of performance as $\mathtt{stride} = 1$ on most datasets.

## A.4 ABLATION STUDY SETUP CHOICES AND EXPLANATIONS

- **Main experiment**: Unlike existing methods that perform dataset-specific hyperparameter tuning to achieve optimal performance, we use the same set of default hyperparameters across all datasets, and later study the effect of each hyperparameter on the metrics in ablation study.

- **Ablation study**: Ablation study in Section 5.3 is conducted on the 30 UEA datasets. For each ablation case, we only change the hyperparameter of interest and keeping other hyperparameters as the default values. A summary of the ablation results are presented in Section 5.3 and detailed results are included in Appendix B.4.

- **Diversity loss**: In the SBM training, we incorporate a regularization (Ma et al., 2020) to encourage the learning of more diverse shapelets (Equation 4). However, the ablation study, we find that the weight of the diversity regularization $\lambda_{\mathrm{div}}$ does not introduce substantial effects to the metrics of interests (see Figure 22). These results suggest that the diversity loss could be omitted, considering its high computational complexity. Nevertheless, we chose to retain it because it is used in other shapelet-based literature (Qu et al., 2024) and may still promote favorable properties for shapelet learning, which would require further investigations.

- **Number of shapelets**: In the ablation study, we consider the number shapelets $K = 10$ and $K = 5$ which are rather compact. Although using large $K$ would test the scalability of SBM, we believe that interpretability favors a compact model with a small number of concepts whenever possible. Using a large $K$ would compromise interpretability by making the SBM more complex and less plausible for human understanding. Furthermore, if the features of a dataset is so complex that require a large $K$ to express, we prefer to leverage the DNN component of InterpGN for such cases.

# B  ADDITIONAL EXPERIMENTAL RESULTS

## B.1  FULL RESULTS ON 30 DATASETS FROM THE UEA ARCHIVE

For the performance of baseline methods, we reproduce STRF, FCN, TimesNet and PatchTST to obtain their performance on the UEA datasets. Performance of DTW and TS2Vec are obtained from Yue et al. (2022). Performance of TST and SVP-T are obtained from Zuo et al. (2023). Performance of ShapeNet, RLPAM, and ShapeConv are obtained from the corresponding paper.

Due to the limitations in model architectures, some method may fail on certain datasets. For example, TimesNet and PatchTST fail on the EigenWorms datasets as the time series exceed their input length limits. For fair comparison, in Table 5, we additionally report the statistics of the methods with "N/A" dropped, keeping the datasets where all the method have a valid result.

Table 5: Accuracy and comparison of the methods on 30 datasets from the UEA archive. The best accuracy for each dataset is marked with bold, the second-best is marked with italic, and the third-best is marked with underline.

| Dataset | Classical | | Deep Learning | | | | | | With Interpretable Features | | | Ours | |
| | STRF | DTW | FCN | TS2Vec | TimesNet | TST | PatchTST | SVP-T | ShapeNet | RLPAM | ShapeConv | SBM | InterpGN |
|---|---|---|---|---|---|---|---|---|---|---|---|---|---|
| ArticularyWordRecognition | 0.917 | 0.987 | 0.990 | 0.987 | 0.953 | 0.983 | 0.938 | *0.993* | 0.987 | 0.923 | **0.994** | *0.993* | 0.991 |
| AtrialFibrillation | 0.267 | 0.200 | 0.333 | 0.200 | 0.507 | 0.200 | 0.467 | 0.400 | 0.400 | **0.733** | *0.521* | 0.453 | 0.333 |
| BasicMotions | 0.925 | 0.975 | **1.000** | 0.975 | 0.965 | 0.965 | 0.560 | **1.000** | **1.000** | **1.000** | 0.997 | **1.000** | **1.000** |
| CharacterTrajectories | 0.849 | 0.989 | **0.998** | 0.995 | 0.980 | N/A | 0.859 | 0.990 | 0.980 | 0.978 | 0.981 | 0.983 | *0.997* |
| Cricket | 0.944 | **1.000** | 0.997 | 0.972 | 0.839 | 0.958 | 0.883 | **1.000** | 0.986 | **1.000** | 0.998 | 0.981 | **1.000** |
| DuckDuckGeese | 0.380 | 0.600 | 0.668 | 0.680 | 0.404 | 0.480 | N/A | *0.700* | **0.725** | *0.700* | 0.648 | 0.384 | 0.480 |
| EigenWorms | 0.672 | 0.618 | 0.759 | 0.847 | N/A | N/A | 0.415 | **0.923** | 0.878 | *0.908* | 0.802 | 0.582 | 0.835 |
| Epilepsy | 0.978 | 0.964 | 0.975 | 0.964 | 0.836 | 0.920 | 0.565 | 0.986 | 0.987 | 0.978 | 0.972 | **0.993** | *0.988* |
| ERing | 0.889 | 0.133 | 0.894 | 0.874 | 0.914 | 0.933 | 0.872 | *0.937* | 0.133 | 0.819 | 0.774 | **0.967** | 0.933 |
| EthanolConcentration | **0.677** | 0.323 | 0.297 | 0.308 | 0.300 | 0.337 | 0.263 | 0.331 | 0.312 | *0.369* | 0.253 | 0.301 | 0.298 |
| FaceDetection | 0.567 | 0.529 | 0.584 | 0.501 | **0.686** | 0.681 | 0.559 | 0.512 | 0.602 | 0.621 | 0.635 | 0.657 | 0.659 |
| FingerMovements | 0.500 | 0.530 | 0.622 | 0.480 | 0.608 | **0.776** | 0.564 | 0.600 | 0.580 | *0.640* | 0.587 | 0.600 | 0.616 |
| HandMovementDirection | 0.419 | 0.231 | 0.446 | 0.338 | 0.416 | *0.608* | 0.568 | 0.392 | 0.338 | **0.635** | 0.413 | 0.473 | 0.459 |
| Handwriting | 0.104 | 0.286 | 0.612 | 0.515 | 0.239 | 0.305 | 0.112 | 0.433 | 0.452 | 0.522 | 0.527 | 0.357 | **0.617** |
| Heartbeat | 0.746 | 0.717 | 0.780 | 0.683 | 0.739 | 0.712 | 0.724 | **0.790** | 0.756 | 0.779 | *0.784* | 0.736 | 0.779 |
| InsectWingbeat | 0.208 | N/A | 0.654 | 0.466 | 0.593 | **0.684** | 0.276 | 0.184 | 0.250 | 0.352 | 0.509 | 0.492 | 0.653 |
| JapaneseVowels | 0.676 | 0.949 | 0.991 | 0.984 | 0.972 | **0.994** | 0.876 | 0.978 | 0.984 | 0.935 | *0.993* | 0.963 | 0.990 |
| Libras | 0.817 | 0.870 | *0.953* | 0.867 | 0.806 | 0.844 | 0.642 | 0.833 | 0.856 | 0.794 | 0.887 | 0.860 | **0.969** |
| LSST | 0.491 | 0.551 | 0.394 | 0.537 | 0.406 | 0.381 | 0.315 | **0.666** | 0.590 | *0.643* | 0.608 | 0.636 | 0.597 |
| MotorImagery | 0.510 | 0.500 | 0.584 | 0.510 | 0.614 | N/A | 0.600 | *0.650* | 0.610 | 0.610 | **0.674** | 0.642 | 0.630 |
| NATOPS | 0.794 | 0.883 | *0.982* | 0.928 | 0.904 | 0.900 | 0.730 | 0.906 | 0.833 | 0.950 | 0.937 | 0.879 | **0.983** |
| PEMS-SF | **0.925** | 0.711 | 0.824 | 0.682 | 0.857 | *0.919* | 0.764 | 0.867 | 0.751 | 0.632 | 0.801 | 0.851 | 0.873 |
| PenDigits | 0.855 | 0.977 | **0.990** | 0.989 | 0.981 | 0.974 | 0.961 | 0.938 | 0.977 | 0.982 | 0.968 | 0.971 | **0.990** |
| PhonemeSpectra | 0.155 | 0.151 | *0.303* | 0.233 | 0.119 | 0.088 | 0.033 | 0.176 | 0.298 | 0.175 | 0.192 | 0.255 | **0.320** |
| RacketSports | 0.842 | 0.803 | **0.903** | 0.855 | 0.855 | 0.829 | 0.738 | 0.842 | 0.882 | 0.868 | 0.863 | 0.897 | *0.901* |
| SelfRegulationSCP1 | 0.846 | 0.775 | 0.902 | 0.812 | 0.895 | **0.925** | 0.835 | 0.884 | 0.782 | 0.802 | 0.858 | 0.864 | *0.917* |
| SelfRegulationSCP2 | 0.489 | 0.539 | 0.577 | 0.578 | 0.559 | 0.589 | 0.559 | 0.600 | 0.578 | **0.632** | *0.624* | 0.522 | 0.578 |
| SpokenArabicDigits | 0.679 | 0.963 | 0.995 | 0.988 | 0.987 | 0.993 | 0.748 | 0.986 | 0.975 | 0.621 | 0.979 | 0.994 | **0.997** |
| StandWalkJump | 0.467 | 0.200 | 0.493 | 0.467 | *0.600* | 0.267 | 0.507 | 0.467 | 0.533 | **0.667** | 0.587 | 0.573 | 0.507 |
| UWaveGestureLibrary | 0.762 | 0.903 | 0.889 | 0.906 | 0.812 | 0.903 | 0.778 | *0.941* | 0.906 | **0.944** | 0.936 | 0.913 | 0.911 |
| Avg. Accuracy | 0.645 | 0.650 | *0.746* | 0.704 | 0.702 | 0.709 | 0.611 | 0.730 | 0.697 | 0.740 | 0.743 | 0.726 | **0.760** |
| Avg. Rank | 9.700 | 9.138 | 5.000 | 7.533 | 7.517 | 6.778 | 10.414 | *5.267* | 6.567 | 5.400 | 5.500 | 5.733 | **3.500** |
| Num. Top-1 | 2 | 1 | 4 | 0 | 1 | 4 | 0 | 5 | 2 | 7 | 2 | 3 | **8** |
| Num. Top-3 | 2 | 1 | *14* | 2 | 3 | 9 | 1 | 11 | 5 | 13 | 10 | 8 | **16** |
| Wins/Draws | 28 | 27 | 23 | 27 | 25 | 18 | 27 | 19 | 25 | 20 | 20 | 20 | - |
| Losses | 2 | 2 | 7 | 3 | 4 | 9 | 2 | 11 | 5 | 10 | 10 | 10 | - |
| Wilcoxon p-value | 0.000 | 0.000 | 0.003 | 0.000 | 0.000 | 0.001 | 0.000 | 0.066 | 0.001 | 0.112 | 0.009 | 0.003 | - |
| **Statistics without N/A** | | | | | | | | | | | | | |
| Avg. Accuracy | 0.669 | 0.646 | *0.749* | 0.705 | 0.710 | 0.719 | 0.622 | 0.738 | 0.699 | 0.747 | 0.748 | 0.748 | **0.768** |
| Avg. Rank | 9.520 | 9.280 | *5.040* | 7.920 | 7.640 | 6.960 | 10.520 | 5.480 | 6.760 | 5.360 | 5.600 | 5.440 | **3.320** |
| Num. Top-1 | 2 | 1 | 3 | 0 | 1 | 3 | 0 | 4 | 1 | 7 | 1 | 3 | **8** |
| Num. Top-3 | 2 | 1 | *12* | 1 | 3 | 8 | 1 | 8 | 3 | 11 | 9 | 7 | **14** |
| Wins/Draws | 23 | 24 | 21 | 24 | 21 | 17 | 23 | 17 | 22 | 17 | 17 | 16 | - |
| Losses | 2 | 1 | 4 | 1 | 4 | 8 | 2 | 8 | 3 | 8 | 8 | 9 | - |
| Wilcoxon p-value | 0.000 | 0.000 | 0.001 | 0.000 | 0.000 | 0.003 | 0.000 | 0.030 | 0.000 | 0.138 | 0.007 | 0.019 | - |

## B.2 ADDITIONAL INTERPRETABILITY ANALYSIS

### B.2.1 FAITHFULNESS

We follow Alvarez Melis & Jaakkola (2018) to asses the faithfulness of shapelets learned by the SBM. Specifically, for each concept (i.e., shapelet predicate $p$), we measure its relevance by calculating the change in the probability of a certain prediction before and after removing it (i.e., set $p = 0$). The faithfulness score is then determined by the correlation between the relevance of a shapelet concept and its assigned importance $w$.

Formally, given a sample $(x_i, y_i = c)$, the SBM first provides the set of shapelet predicates $\mathbf{p}_i \in \mathbb{R}^{|S|} = [p_{i,1}, \ldots, p_{i,|\mathcal{S}|}]$ and their corresponding classifier weights $\mathbf{w} \in \mathbb{R}^{|S| \times C} = [w_{1,c}, \ldots, w_{|\mathcal{S}|,c} \mid c \in \{1, \ldots, C\}]$. Then, the predicted class probabilities are computed by $\hat{\mathbf{r}}_i = \texttt{softmax}(\mathbf{w}^\top \mathbf{p}_i)$ and $\hat{\mathbf{r}}_i = [\hat{r}_1, \ldots, \hat{r}_C]$. Then, we have

$$\Delta_i^{(k)} = \hat{r}_{i,c} - \hat{r}_{i,c}^{(k)}, \text{ where } \hat{\mathbf{r}}_i^{(k)} = \texttt{softmax}(\mathbf{w}^\top \mathbf{p}_i^{(k)}),$$
$$\mathbf{p}_i^{(k)} = [p_{i,j} \mid j = \{1, \ldots, |S|, p_{i,j} = 0 \text{ if } j = k\}]. \tag{10}$$

Here $\Delta_i^{(k)}$ measures the changes in the predicted class probability with the absence of shapelet concept $s_k$. The faithfulness of the sample $x_i$ is then quantified by the correlation coefficient between the vectors of relevance values $\Delta_i = [\Delta_i^{(1)}, \ldots, \Delta_i^{(|S|)}]$ and the importance weights of the predicted class $\mathbf{w}_c$:

$$\text{Faithfulness}(x_i) = \rho(\mathbf{w}_i, \Delta_i). \tag{11}$$

where $\rho$ denotes the Pearson correlation coefficient. Aggregating Faithfulness($x_i$) across all $(x_i, y_i) \in D$, we generate the box plots shown in Figure 7. On most datasets, regardless of the varying performance levels of the SBM, we observe consistently high faithfulness estimates (i.e., greater than 0.5).

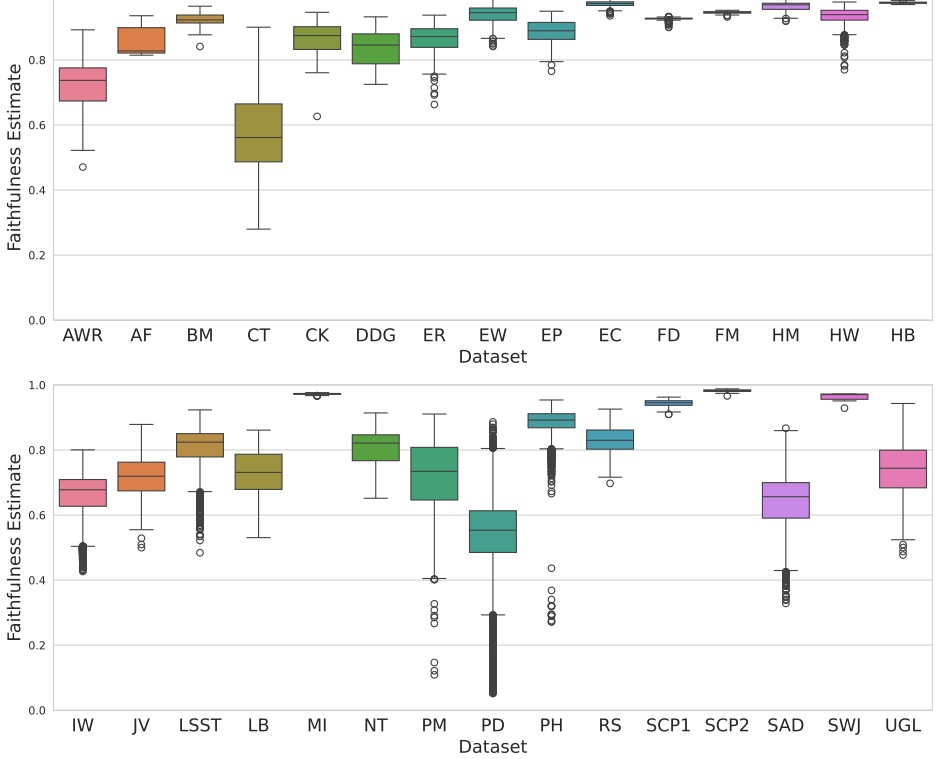

Figure 7: Faithfulness estimate: aggregated correlation statistics between shapelet concept relevance and importance.

### B.2.2 VISUALIZATION OF EXPLANATIONS

**Global explanations** . We visualize the global explanations of the UWaveGestureLibrary, BasicMotions, SelfRegulationSCP1, and SelfRegulationSCP2 datasets. We choose these datasets since their dimensions are favored for visualizations (i.e., $M$ and $C$ are not too high). UWaveGestureLibrary and BasicMotions are representitives for human motion data; SelfRegulationSCP1 and SelfRegulationSCP2 are representatives for EEG data. In each figure, for each category, we visualize the time-series samples (left, in gray), the top-5 positively relevant shapelets (middle, in blue), and the top-5 negatively relevant shapelets (right, in red).

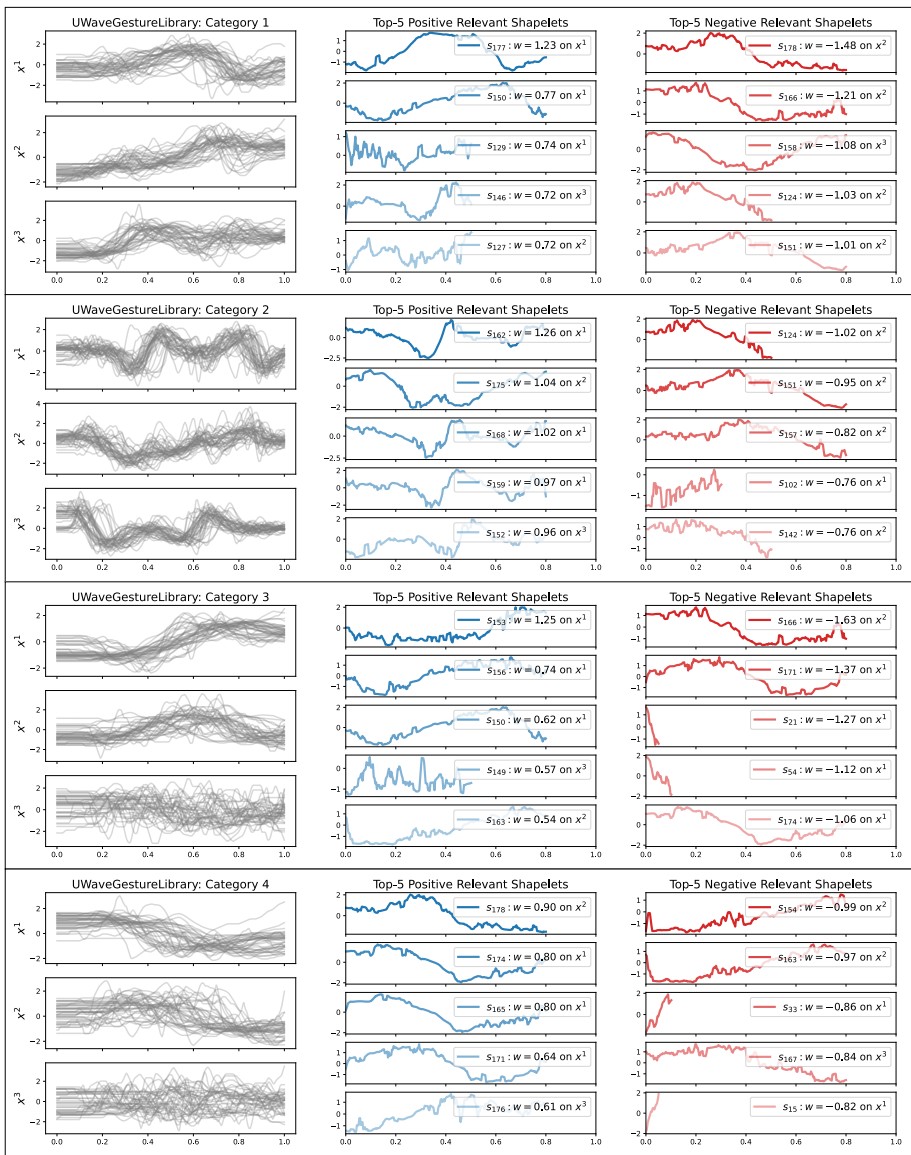

Figure 8: Global explanations of the SBM on the UWaveGestureLibrary dataset. (Part 1)

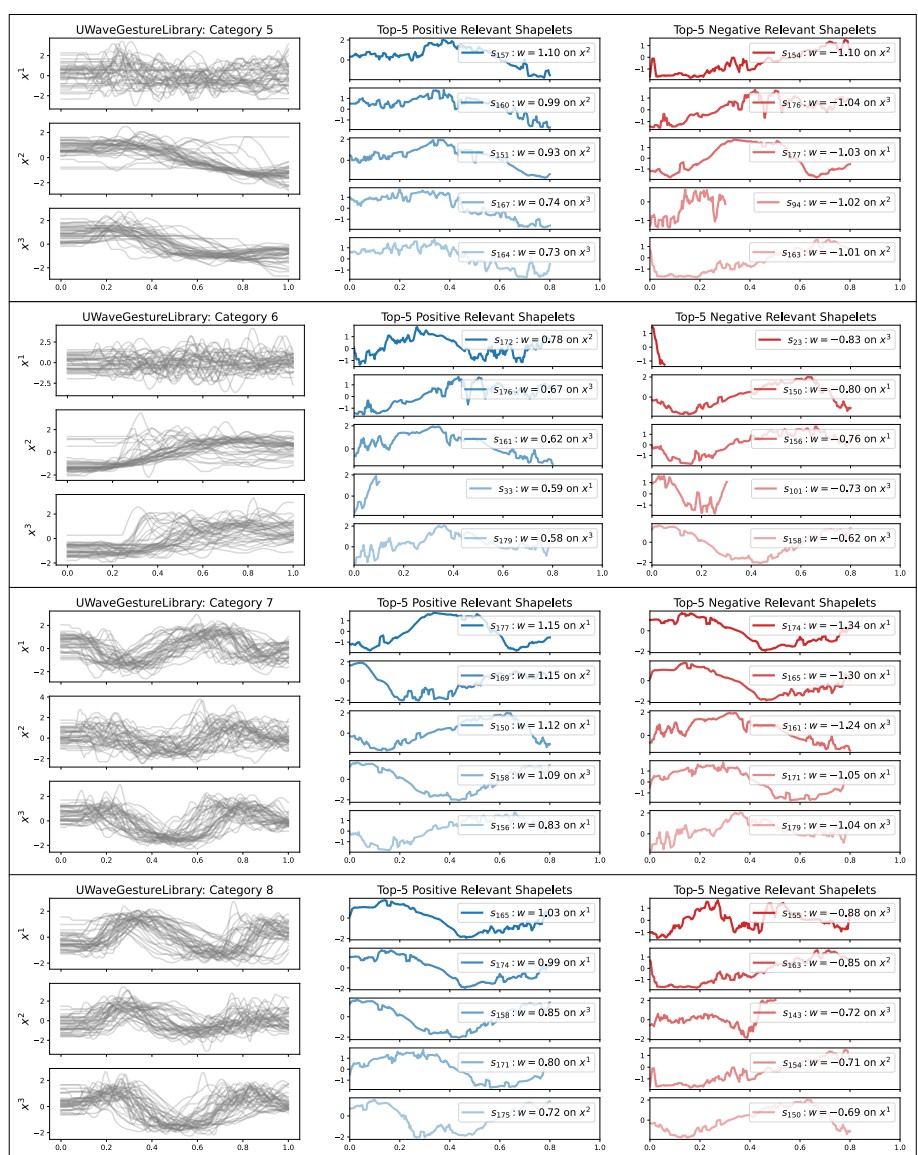

Figure 9: Global explanations of the SBM on the UWaveGestureLibrary dataset. (Part 2)

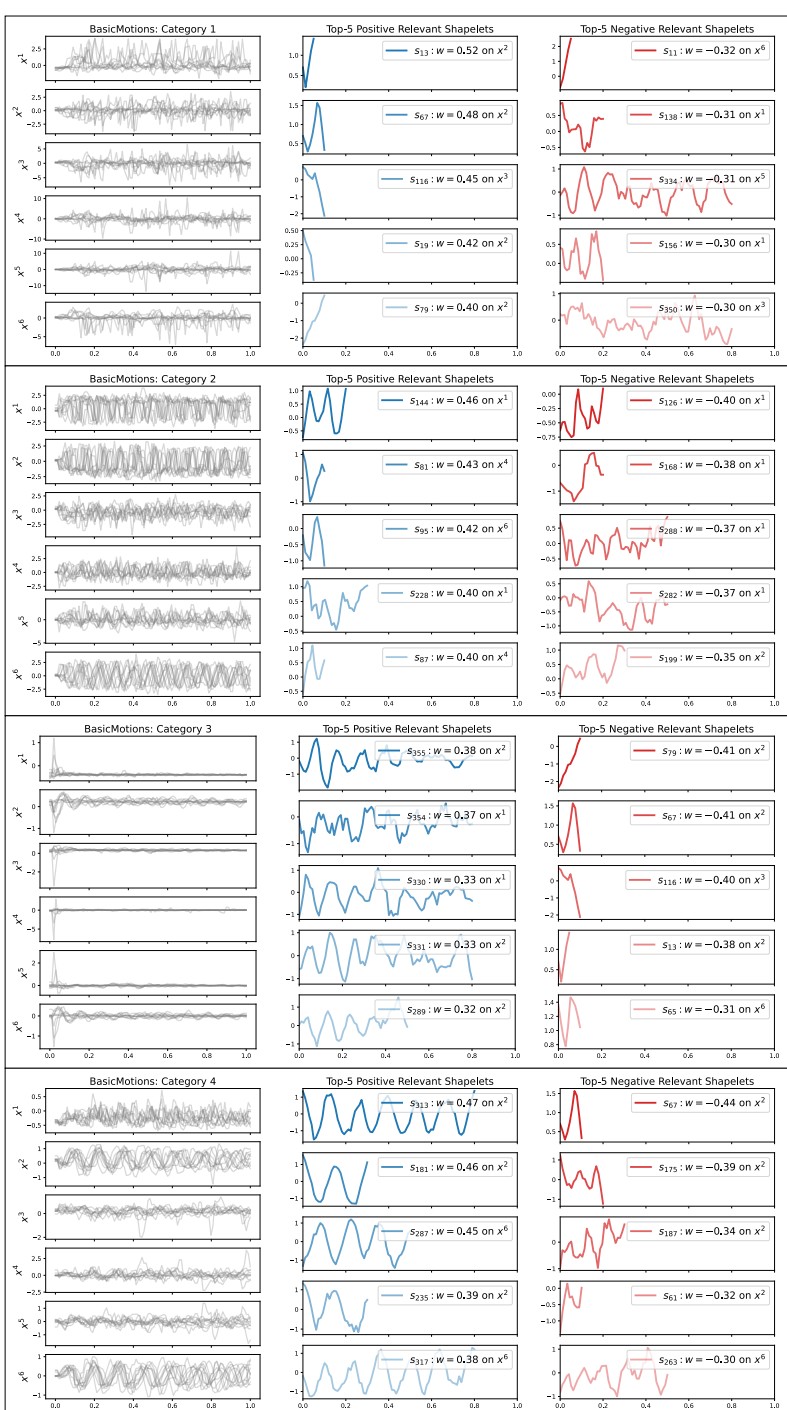

Figure 10: Global explanations of the SBM on the BasicMotions dataset.

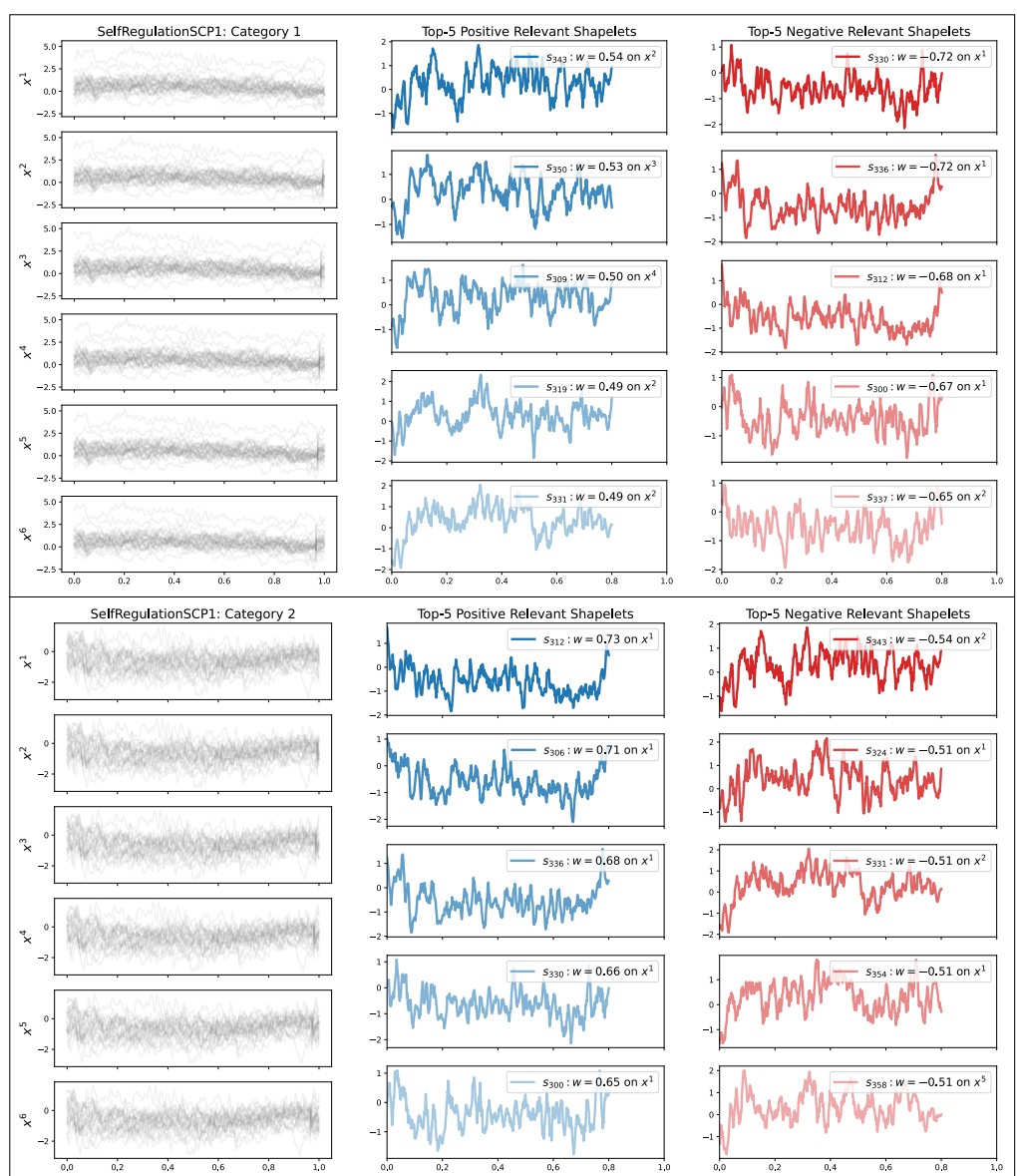

Figure 11: Global explanations of the SBM on the SelfRegulationSCP1 dataset. For clarity, we visualize 30 time-series samples.

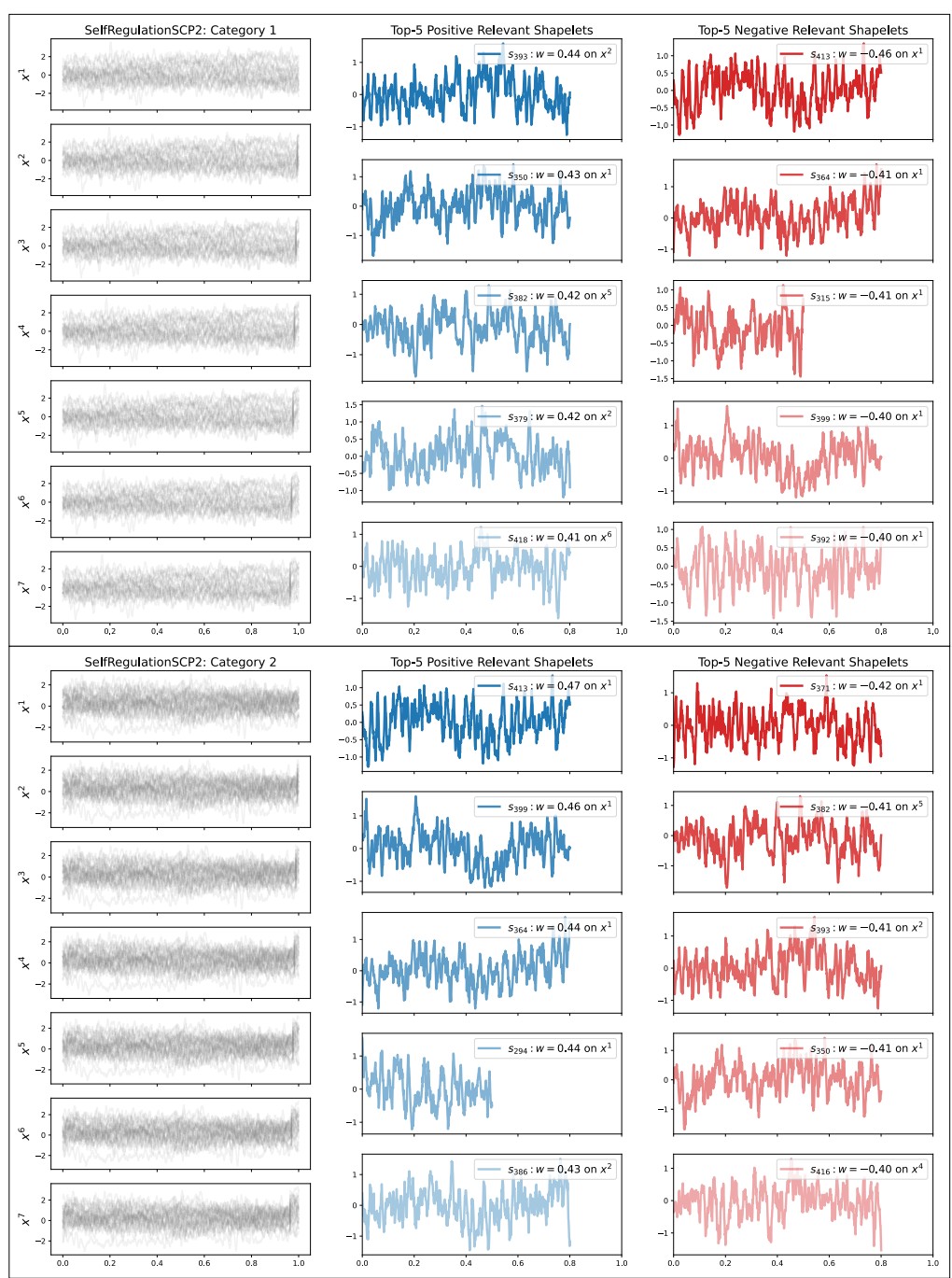

Figure 12: Global explanations of the SBM on the SelfRegulationSCP2 dataset. For clarity, we visualize 30 time-serie samples.

**Local explanations.** We visualize the local explanations for the four datasets. In each Figure, a sub-figure visualizes a time-series sample (in gray) along with the top-5 most important shapelets. The color of each shapelet indicates its importance, with blue representing the highest importance, followed by orange, green, red, and purple for progressively lower importance.

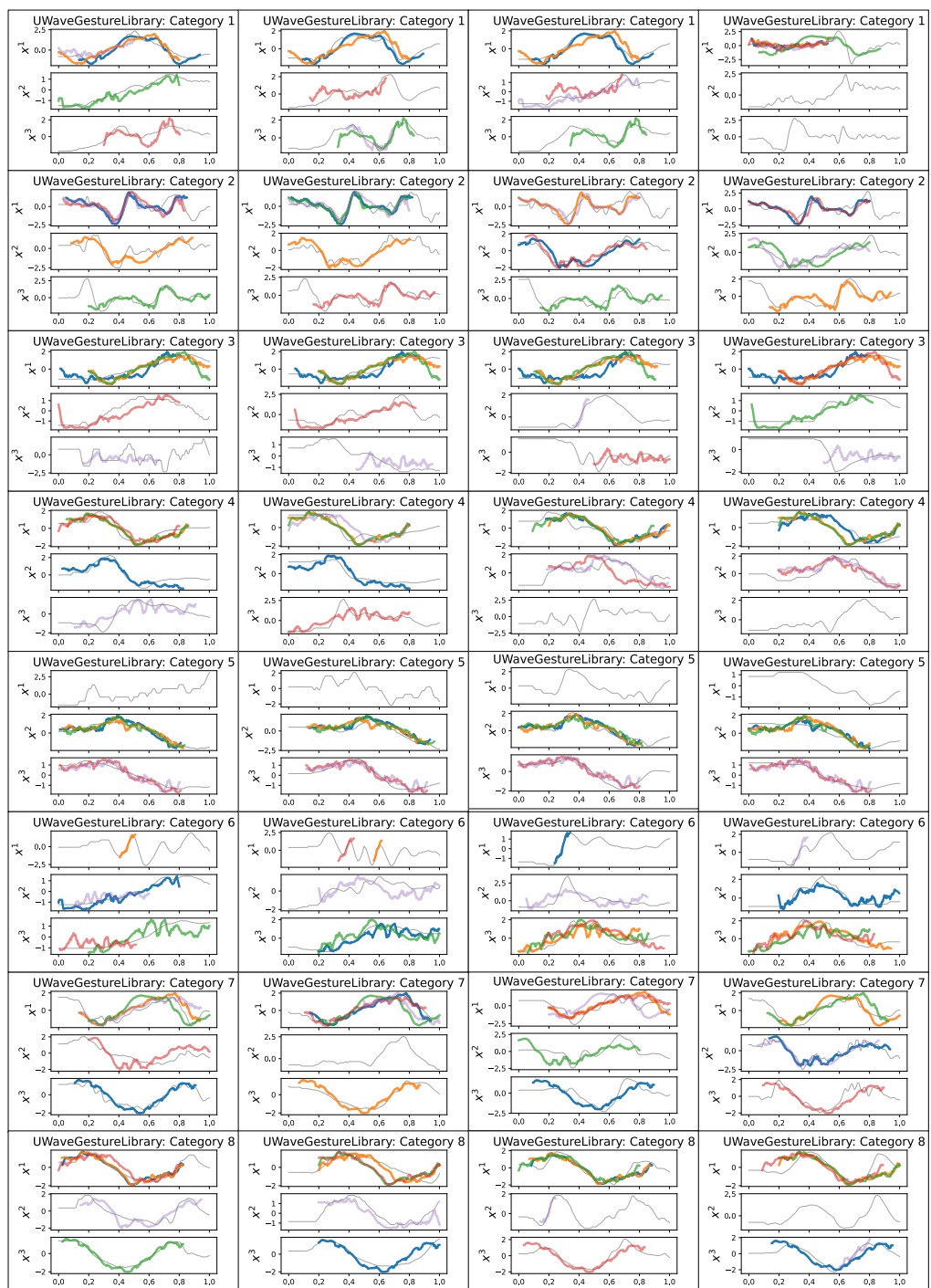

Figure 13: Local explanations of the SBM on the UWaveGestureLibrary dataset.

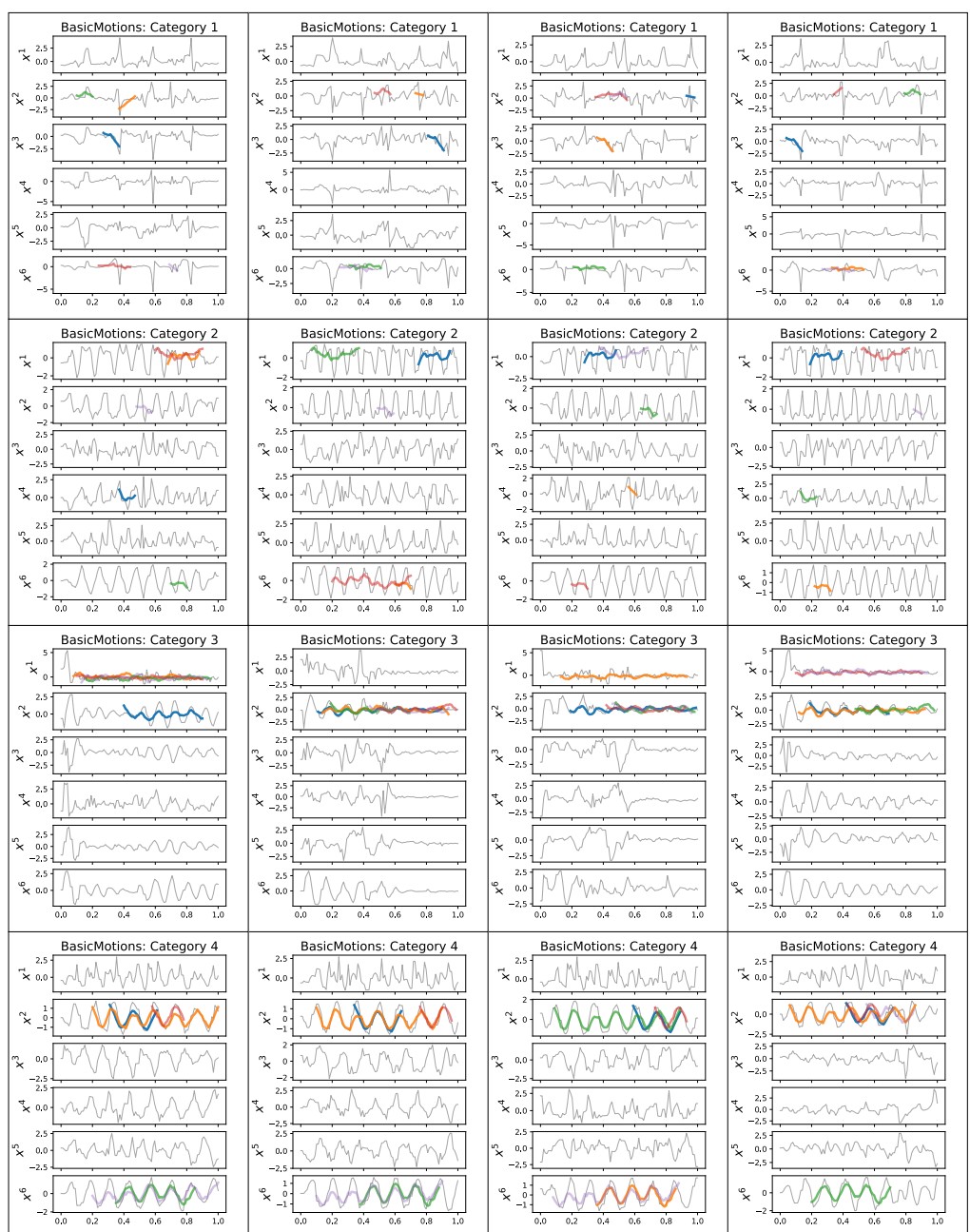

Figure 14: Local explanations of the SBM on the BasicMotions dataset.

The local explanations reflect the limitation we discussed in Section 6, where explanations for certain categories may appear counterfactual. For example, in the BasicMotions dataset, the most important shapelets for Category 1 may not represent the most characteristic features of that category. Despite this, the SBM can still make accurate predictions due to the strong feature representation for other categories. This issue becomes more pronounced in datasets with two categories, such as SelfRegulationSCP1 and SelfRegulationSCP2. These findings suggest that for binary classification tasks, using a single binary classifier with one unified "rule" might be a more effective and reasonable design, rather than assigning a separate "rule" to each class.

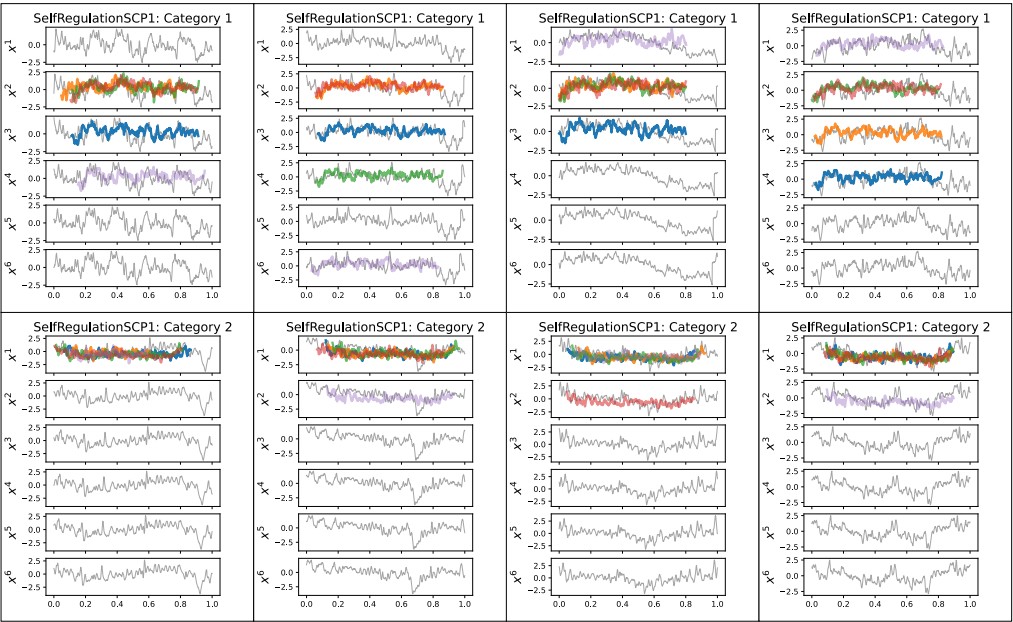

Figure 15: Local explanations of the SBM on the SelfRegulationSCP1 dataset.

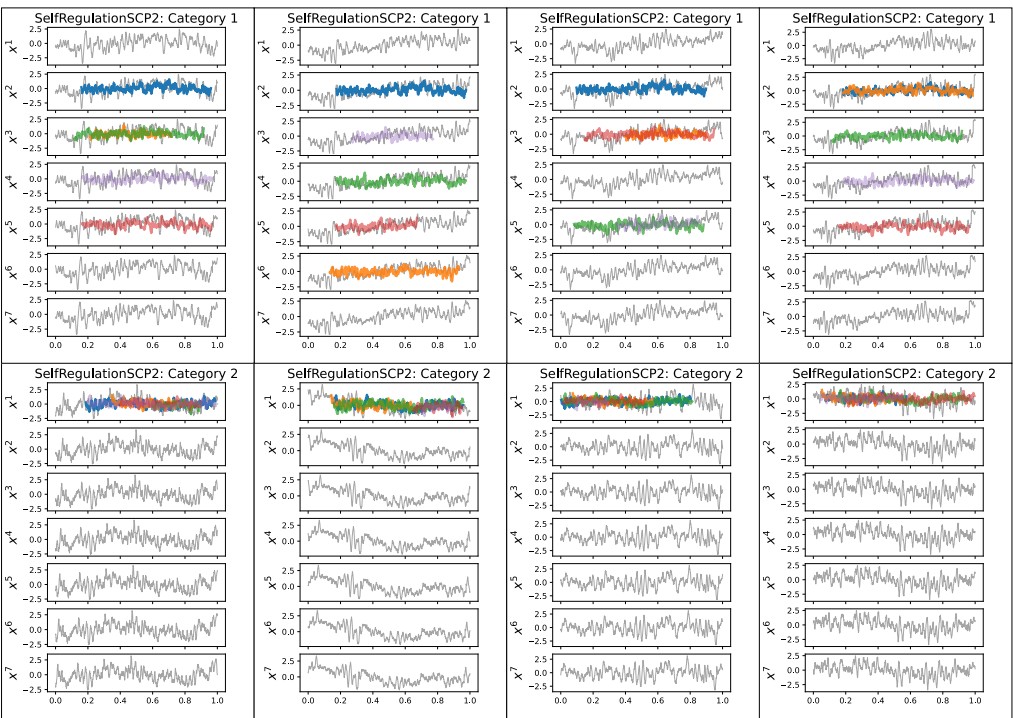

Figure 16: Local explanations of the SBM on the SelfRegulationSCP2 dataset.

## B.3 BEHAVIOR OF THE INTERPRETABILITY GATING FUNCTION

### B.3.1 DISTRIBUTION OF THE GATING VALUES

In Figure 17, we present the relationship between values of $\eta(x_i)$ and predictions of InterpGN to further study the behavior of the proposed interpretability gating function. We conclude that InterpGN's predictions is consistent with SBM's predictions when $\eta(x_i)$ is high, preserving interpretability for those samples. The DNN makes corrections when $\eta(x_i)$ is low to improve performance. Note that there are cases where the DNN changes SBM's predictions to incorrect, suggesting that InterpGN's performance can be further improved using more advanced DNNs.

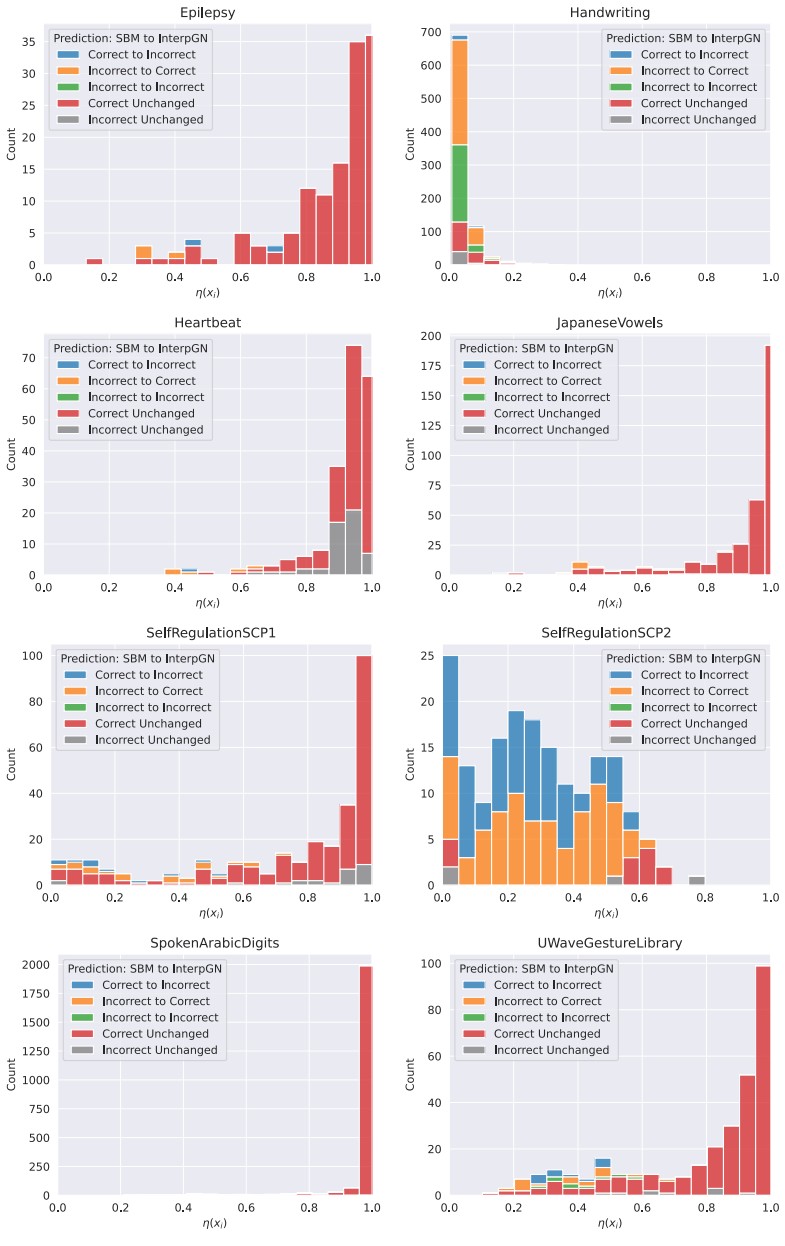

Figure 17: Histogram of values $\eta(x_i)$, and the prediction consistency between the interpretable expert and InterpGN.

### B.3.2 TRAINING CURVES OF INTERPGN

In Figure 18 and Figure 19, we visualize the loss and $\eta$ during the training of InterpGN. The figures present how the loss function design $\mathcal{L}_{\text{hybrid}} = \beta \mathcal{L}_{\text{int}} + \hat{\mathcal{L}}_{\text{ce}}$ could encourage the use of SBM during training, preventing the InterpGN from collapsing into a pure DNN when possible.

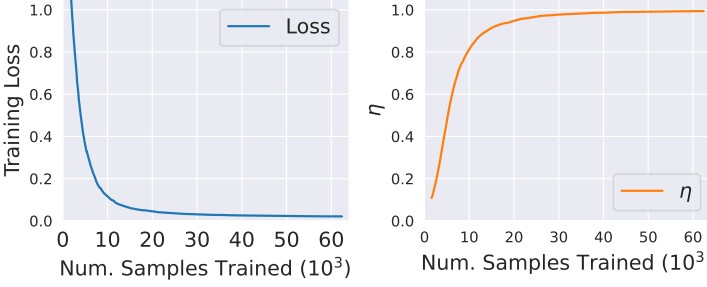

Figure 18: Training curves for InterpGN on the BasicMotions dataset: (left) $L_{\text{hybrid}}$ and (right) average $\eta$.

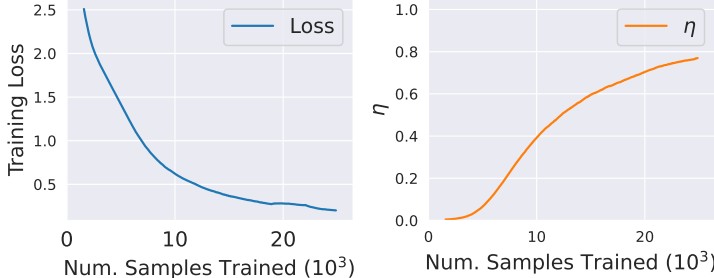

Figure 19: Training curves for InterpGN on the UWaveGestureLibrary dataset: (left) $L_{\text{hybrid}}$ and (right) average $\eta$.

## B.4 ABLATION STUDY RESULTS

Table 6: Number of shapelet

| model | $K$ | accu. | quality | $|w| > 1$ | $|w| > 0.5$ | $|w| > 0.1$ | Gini($|w|$) | $\eta$ |
|---|---|---|---|---|---|---|---|---|
| SBM | 5 | 0.720 | 0.103 | 0.050 | 0.144 | 0.523 | 0.460 | - |
| SBM | 10 | 0.726 | 0.074 | 0.026 | 0.084 | 0.412 | 0.467 | - |
| InterpGN | 5 | 0.761 | 0.079 | 0.034 | 0.109 | 0.429 | 0.422 | 0.405 |
| InterpGN | 10 | 0.760 | 0.061 | 0.014 | 0.060 | 0.343 | 0.420 | 0.441 |

Table 7: $\beta$ schedule of InterpGN

| $\beta$ | $K$ | accu. | quality | $|w| > 1$ | $|w| > 0.5$ | $|w| > 0.1$ | Gini($|w|$) | $\eta$ |
|---|---|---|---|---|---|---|---|---|
| constant | 5 | 0.761 | 0.079 | 0.034 | 0.109 | 0.429 | 0.422 | 0.405 |
| constant | 10 | 0.760 | 0.061 | 0.014 | 0.060 | 0.343 | 0.420 | 0.441 |
| cosine | 5 | 0.759 | 0.077 | 0.031 | 0.096 | 0.427 | 0.422 | 0.374 |
| cosine | 10 | 0.761 | 0.059 | 0.015 | 0.060 | 0.334 | 0.424 | 0.435 |

Table 8: Predicate types in SBM[2]

| Predicate | $K$ | accu. | quality | $|w| > 1$ | $|w| > 0.5$ | $|w| > 0.1$ |
|---|---|---|---|---|---|---|
| RBF | 5 | 0.728 | 0.101 | 0.051 | 0.141 | 0.513 |
| RBF | 10 | 0.734 | 0.073 | 0.027 | 0.083 | 0.405 |
| Linear | 5 | 0.718 | 0.171 | 0.084 | 0.181 | 0.499 |
| Linear | 10 | 0.722 | 0.121 | 0.054 | 0.125 | 0.433 |

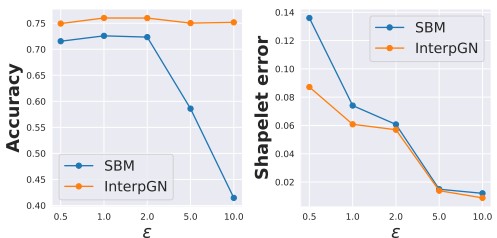

Figure 20: **Ablation on** $\epsilon$

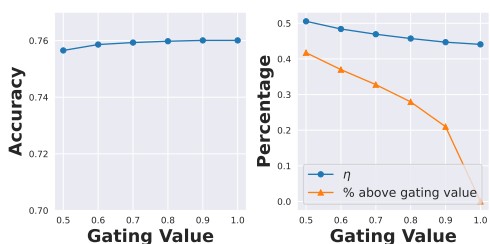

Figure 21: **Ablation on** $\underline{\eta}$

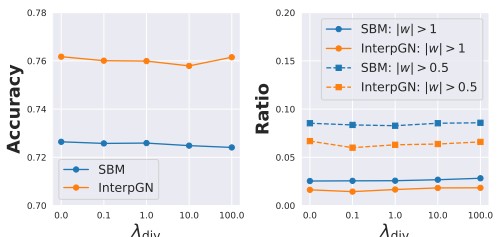

Figure 22: **Ablation on** $\lambda_{\mathbf{div}}$

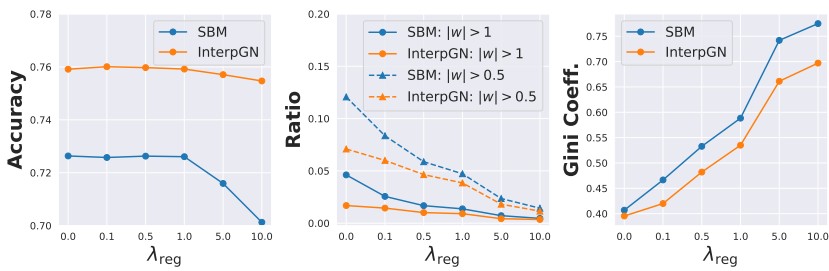

Figure 23: **Ablation on** $\lambda_{\mathbf{reg}}$

Table 9: Full results and variations on number of shapelets $K$

| model | $K$ | accu. | quanlity | $|w| > 1$ | $|w| > 0.5$ | $|w| > 0.1$ | $\eta$ |
|---|---|---|---|---|---|---|---|
| SBM | 5 | $0.72 \pm 0.014$ | $0.103 \pm 0.023$ | $0.05 \pm 0.017$ | $0.144 \pm 0.051$ | $0.523 \pm 0.109$ | - |
| SBM | 10 | $0.726 \pm 0.011$ | $0.074 \pm 0.014$ | $0.026 \pm 0.008$ | $0.084 \pm 0.022$ | $0.412 \pm 0.085$ | - |
| InterpGN | 5 | $0.761 \pm 0.012$ | $0.079 \pm 0.019$ | $0.034 \pm 0.011$ | $0.109 \pm 0.036$ | $0.429 \pm 0.105$ | $0.405 \pm 0.091$ |
| InterpGN | 10 | $0.76 \pm 0.01$ | $0.061 \pm 0.014$ | $0.014 \pm 0.006$ | $0.06 \pm 0.024$ | $0.343 \pm 0.092$ | $0.441 \pm 0.112$ |

---

[3]Results in this table are averaged over 28 datasets since the models with linear predicates fail on the datasets EigenWorms and MotorImagery

Table 10: Full results and variations on $\beta$ schedule in InterpGN

| $\beta$ schedule | $K$ | accu. | quanlity | $|w| > 1$ | $|w| > 0.5$ | $|w| > 0.1$ | $\eta$ |
|---|---|---|---|---|---|---|---|
| constant | 5 | $0.761 \pm 0.012$ | $0.079 \pm 0.019$ | $0.034 \pm 0.011$ | $0.109 \pm 0.036$ | $0.429 \pm 0.105$ | $0.405 \pm 0.091$ |
| constant | 10 | $0.76 \pm 0.01$ | $0.061 \pm 0.014$ | $0.014 \pm 0.006$ | $0.06 \pm 0.024$ | $0.343 \pm 0.092$ | $0.441 \pm 0.112$ |
| cosine | 5 | $0.759 \pm 0.016$ | $0.077 \pm 0.019$ | $0.031 \pm 0.014$ | $0.096 \pm 0.028$ | $0.427 \pm 0.103$ | $0.374 \pm 0.099$ |
| cosine | 10 | $0.761 \pm 0.011$ | $0.059 \pm 0.014$ | $0.015 \pm 0.007$ | $0.06 \pm 0.02$ | $0.334 \pm 0.09$ | $0.435 \pm 0.1$ |

Table 11: Full results and variations on $\epsilon$

| model | $\epsilon$ | accu. | quanlity | $|w| > 1$ | $|w| > 0.5$ | $|w| > 0.1$ | $\eta$ |
|---|---|---|---|---|---|---|---|
| SBM | 0.5 | $0.715 \pm 0.014$ | $0.136 \pm 0.03$ | $0.033 \pm 0.008$ | $0.089 \pm 0.028$ | $0.383 \pm 0.091$ | - |
| SBM | 1.0 | $0.726 \pm 0.011$ | $0.074 \pm 0.014$ | $0.026 \pm 0.008$ | $0.084 \pm 0.022$ | $0.412 \pm 0.085$ | - |
| SBM | 2.0 | $0.723 \pm 0.01$ | $0.061 \pm 0.011$ | $0.029 \pm 0.008$ | $0.111 \pm 0.034$ | $0.465 \pm 0.095$ | - |
| SBM | 5.0 | $0.586 \pm 0.029$ | $0.015 \pm 0.004$ | $0.031 \pm 0.009$ | $0.063 \pm 0.016$ | $0.138 \pm 0.022$ | - |
| SBM | 10.0 | $0.415 \pm 0.026$ | $0.012 \pm 0.004$ | $0.023 \pm 0.008$ | $0.042 \pm 0.014$ | $0.072 \pm 0.011$ | - |
| InterpGN | 0.5 | $0.749 \pm 0.015$ | $0.087 \pm 0.025$ | $0.016 \pm 0.004$ | $0.05 \pm 0.018$ | $0.251 \pm 0.093$ | $0.31 \pm 0.114$ |
| InterpGN | 1.0 | $0.76 \pm 0.01$ | $0.061 \pm 0.014$ | $0.014 \pm 0.006$ | $0.06 \pm 0.024$ | $0.343 \pm 0.092$ | $0.441 \pm 0.112$ |
| InterpGN | 2.0 | $0.76 \pm 0.009$ | $0.057 \pm 0.01$ | $0.023 \pm 0.009$ | $0.097 \pm 0.028$ | $0.431 \pm 0.094$ | $0.488 \pm 0.098$ |
| InterpGN | 5.0 | $0.75 \pm 0.012$ | $0.014 \pm 0.004$ | $0.029 \pm 0.008$ | $0.054 \pm 0.016$ | $0.136 \pm 0.028$ | $0.263 \pm 0.059$ |
| InterpGN | 10.0 | $0.752 \pm 0.012$ | $0.009 \pm 0.003$ | $0.024 \pm 0.007$ | $0.04 \pm 0.014$ | $0.072 \pm 0.017$ | $0.118 \pm 0.038$ |

Table 12: Full results and variations on $\lambda_{\text{reg}}$

| model | $\lambda_{\text{reg}}$ | accu. | quanlity | $|w| > 1$ | $|w| > 0.5$ | $|w| > 0.1$ | $\eta$ |
|---|---|---|---|---|---|---|---|
| SBM | 0.0 | $0.726 \pm 0.01$ | $0.113 \pm 0.023$ | $0.046 \pm 0.01$ | $0.121 \pm 0.03$ | $0.487 \pm 0.09$ | - |
| SBM | 0.1 | $0.726 \pm 0.011$ | $0.074 \pm 0.014$ | $0.026 \pm 0.008$ | $0.084 \pm 0.022$ | $0.412 \pm 0.085$ | - |
| SBM | 0.5 | $0.726 \pm 0.01$ | $0.059 \pm 0.012$ | $0.017 \pm 0.006$ | $0.059 \pm 0.017$ | $0.338 \pm 0.073$ | - |
| SBM | 1.0 | $0.726 \pm 0.011$ | $0.048 \pm 0.007$ | $0.014 \pm 0.004$ | $0.047 \pm 0.011$ | $0.259 \pm 0.064$ | - |
| SBM | 5.0 | $0.716 \pm 0.011$ | $0.022 \pm 0.004$ | $0.007 \pm 0.002$ | $0.024 \pm 0.008$ | $0.1 \pm 0.028$ | - |
| SBM | 10.0 | $0.701 \pm 0.012$ | $0.014 \pm 0.003$ | $0.005 \pm 0.001$ | $0.014 \pm 0.004$ | $0.058 \pm 0.015$ | - |
| InterpGN | 0.0 | $0.759 \pm 0.013$ | $0.067 \pm 0.02$ | $0.017 \pm 0.011$ | $0.071 \pm 0.031$ | $0.381 \pm 0.093$ | $0.463 \pm 0.114$ |
| InterpGN | 0.1 | $0.76 \pm 0.01$ | $0.061 \pm 0.014$ | $0.014 \pm 0.006$ | $0.06 \pm 0.024$ | $0.343 \pm 0.092$ | $0.441 \pm 0.112$ |
| InterpGN | 0.5 | $0.76 \pm 0.011$ | $0.049 \pm 0.01$ | $0.01 \pm 0.004$ | $0.046 \pm 0.016$ | $0.271 \pm 0.081$ | $0.44 \pm 0.104$ |
| InterpGN | 1.0 | $0.759 \pm 0.011$ | $0.042 \pm 0.009$ | $0.009 \pm 0.004$ | $0.038 \pm 0.01$ | $0.221 \pm 0.069$ | $0.432 \pm 0.096$ |
| InterpGN | 5.0 | $0.757 \pm 0.01$ | $0.02 \pm 0.004$ | $0.004 \pm 0.002$ | $0.018 \pm 0.006$ | $0.09 \pm 0.036$ | $0.358 \pm 0.093$ |
| InterpGN | 10.0 | $0.755 \pm 0.013$ | $0.013 \pm 0.003$ | $0.004 \pm 0.002$ | $0.011 \pm 0.005$ | $0.047 \pm 0.02$ | $0.299 \pm 0.092$ |

Table 13: Full results and variations on $\lambda_{\text{div}}$

| model | $\lambda_{\text{div}}$ | accu. | quanlity | $|w| > 1$ | $|w| > 0.5$ | $|w| > 0.1$ | $\eta$ |
|---|---|---|---|---|---|---|---|
| SBM | 0.0 | $0.726 \pm 0.011$ | $0.072 \pm 0.013$ | $0.025 \pm 0.005$ | $0.085 \pm 0.019$ | $0.413 \pm 0.093$ | - |
| SBM | 0.1 | $0.726 \pm 0.011$ | $0.074 \pm 0.014$ | $0.026 \pm 0.008$ | $0.084 \pm 0.022$ | $0.412 \pm 0.085$ | - |
| SBM | 1.0 | $0.726 \pm 0.01$ | $0.079 \pm 0.016$ | $0.026 \pm 0.009$ | $0.083 \pm 0.022$ | $0.417 \pm 0.089$ | - |
| SBM | 10.0 | $0.725 \pm 0.01$ | $0.089 \pm 0.017$ | $0.027 \pm 0.009$ | $0.085 \pm 0.024$ | $0.42 \pm 0.079$ | - |
| SBM | 100.0 | $0.724 \pm 0.011$ | $0.097 \pm 0.019$ | $0.028 \pm 0.009$ | $0.086 \pm 0.022$ | $0.406 \pm 0.091$ | - |
| InterpGN | 0.0 | $0.762 \pm 0.012$ | $0.062 \pm 0.014$ | $0.016 \pm 0.007$ | $0.067 \pm 0.021$ | $0.362 \pm 0.102$ | $0.451 \pm 0.112$ |
| InterpGN | 0.1 | $0.76 \pm 0.01$ | $0.061 \pm 0.014$ | $0.014 \pm 0.006$ | $0.06 \pm 0.024$ | $0.343 \pm 0.092$ | $0.441 \pm 0.112$ |
| InterpGN | 1.0 | $0.76 \pm 0.011$ | $0.065 \pm 0.013$ | $0.017 \pm 0.006$ | $0.063 \pm 0.021$ | $0.338 \pm 0.084$ | $0.443 \pm 0.107$ |
| InterpGN | 10.0 | $0.758 \pm 0.011$ | $0.073 \pm 0.016$ | $0.018 \pm 0.009$ | $0.064 \pm 0.027$ | $0.331 \pm 0.086$ | $0.437 \pm 0.109$ |
| InterpGN | 100.0 | $0.762 \pm 0.011$ | $0.081 \pm 0.018$ | $0.018 \pm 0.008$ | $0.066 \pm 0.023$ | $0.331 \pm 0.099$ | $0.437 \pm 0.113$ |

## C PRELIMINARY RESULTS ON EXTENDED DESIGNS

We further provide preliminary results on extended designs for SBM and InterpGN, where we hope they could provide insights for future research. For simplicity, we experiment with a subset of the UEA archive containing 10 datasets that are commonly used in evaluating deep learning methods (Wu et al., 2023).

### C.1 ARCHITECTURE OF THE DNN EXPERT IN INTERPGN

Table 14: Ablation on the architecture of the DNN expert in InterpGN. The best performance for each dataset is highlighted with bold.

| DNN Expert of InterpGN: | FCN | | Transformer | | PatchTST | | TimesNet | |
|---|---|---|---|---|---|---|---|---|
| Dataset | accu. | $\eta$ | accu. | $\eta$ | accu. | $\eta$ | accu. | $\eta$ |
| EthanolConcentration | 0.298 | 0.008 | **0.320** | 0.236 | 0.276 | 0.833 | 0.302 | 0.401 |
| FaceDetection | 0.663 | 0.709 | 0.657 | 0.568 | 0.652 | 0.756 | **0.670** | 0.517 |
| Handwriting | **0.612** | 0.022 | 0.127 | 0.545 | 0.115 | 0.266 | 0.214 | 0.024 |
| Heartbeat | **0.779** | 0.246 | 0.735 | 0.671 | 0.729 | 0.954 | 0.740 | 0.576 |
| JapaneseVowels | **0.991** | 0.758 | 0.970 | 0.920 | 0.919 | 0.871 | 0.974 | 0.831 |
| PEMS-SF | **0.861** | 0.480 | 0.800 | 0.671 | 0.206 | 0.980 | 0.834 | 0.560 |
| SelfRegulationSCP1 | **0.917** | 0.428 | 0.878 | 0.448 | 0.851 | 0.407 | 0.902 | 0.437 |
| SelfRegulationSCP2 | **0.580** | 0.036 | 0.559 | 0.700 | 0.576 | 0.587 | 0.578 | 0.363 |
| SpokenArabicDigits | **0.998** | 0.959 | 0.993 | 0.984 | 0.991 | 0.985 | 0.992 | 0.969 |
| UWaveGestureLibrary | **0.909** | 0.646 | 0.803 | 0.365 | 0.791 | 0.295 | 0.829 | 0.205 |

### C.2 SHAPELET DISTANCE MEASUREMENTS

In this paper, as defined in Equation 1, we follow the conventional definition of shapelet distance and use Euclidean distance, but the shapelet distance can be general and replaced by other metrics. Here we explore two alternative distance measures: cosine similarity and Pearson correlation. These metrics may offer favorable properties compared to Euclidean distance in some cases. For instance, cosine similarity can capture shapelets that are invariant to changes in offset. Table 15 presents the classification accuracy of SBM and InterpGN variants utilizing these three distance measures. The results indicate that cosine similarity and Pearson correlation can yield better performance on certain datasets, but no metric can be universally better than others. This suggests that the choice of shapelet distance metric could possibly serve as a hyperparameter requiring dataset-specific tuning depending on the needs. For example, motion datasets such as UWaveGestureLibrary may benefit from cosine similarity due to the presence of offset-invariant shapelet features. In Figure 24, we visualize an example of local explanations of SBM using euclidean distance and cosine similarity as the shapelet distance measurement, which suggests that they both could learn high-quality shapelets.

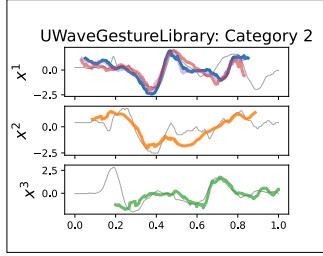
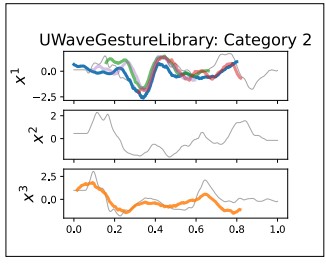

(a) SBM with Euclidean Distance     (b) SBM with Cosine Similarity

Figure 24: Local explanations of SBM using (a) euclidean distance and (b) cosine similarity as the shapelet distance measurement.

Table 15: Classification accuracy of the SBM using three shapelet distance metrics. The best accuracy for each dataset is marked with bold. The training of SBM using Pearson correlation fails on some datasets due to infinite gradients.

| Model: | SBM | | | InterpGN | | |
|---|---|---|---|---|---|---|
| Distance metric: | Euclidean | Cosine | Pearson | Euclidean | Cosine | Pearson |
| EthanolConcentration | 0.301 | 0.297 | **0.304** | 0.298 | **0.304** | 0.291 |
| FaceDetection | 0.657 | 0.680 | **0.682** | 0.659 | **0.683** | 0.682 |
| Handwriting | 0.357 | **0.395** | 0.042 | **0.617** | 0.613 | 0.042 |
| Heartbeat | 0.736 | 0.738 | **0.740** | 0.779 | **0.789** | 0.779 |
| JapaneseVowels | 0.963 | **0.966** | 0.084 | **0.990** | 0.989 | 0.084 |
| PEMS-SF | **0.850** | 0.830 | 0.173 | 0.853 | 0.853 | 0.173 |
| SelfRegulationSCP1 | **0.864** | 0.857 | 0.855 | **0.917** | 0.900 | 0.901 |
| SelfRegulationSCP2 | 0.522 | 0.553 | **0.556** | 0.578 | 0.578 | **0.579** |
| SpokenArabicDigits | **0.994** | 0.991 | 0.100 | **0.997** | 0.995 | 0.100 |
| UWaveGestureLibrary | 0.913 | **0.946** | 0.125 | 0.911 | **0.936** | 0.125 |

## C.3 CLASSIFIER ARCHITECTURE IN SBM

In this paper, we focus on using a linear classifier with shapelet predicates, in line with existing shapelet-based methods. While the linear classifier is often preferred for its interpretability, its simplicity introduces limitations such as neglect of the interaction/correlation among shapelets, as discussed in Section 6. To mitigate the issues, we present two additional classifier architectures: a bi-linear classifier and an attention-based classifier. Here, for simplicity, we denote the set of shapelets predicates as $\mathbf{p}_i = [p_{i,1}, \ldots, p_{i,|\mathcal{S}|}]$. Specifically, they are formulated as

- Bi-linear classifier:

$$r_{i,c} = \sum_{k=1}^{|\mathcal{S}|} w_k \, p_{i,k} + \sum_{k=1}^{|\mathcal{S}|} \sum_{j=1}^{|\mathcal{S}|} \mathbb{1}(k \neq j) w_{j,k} \, p_{i,k} \, p_{i,j}. \tag{12}$$

- Attention-based classifier:

$$r_{i,c} = \sum_{k=1}^{|\mathcal{S}|} w_k \cdot \texttt{attention}(Q(\mathbf{p}_i) + \texttt{PE}, K(\mathbf{p}_i) + \texttt{PE}, p_{i,k}), \tag{13}$$

where `attention` is the scaled dot-product attention, `PE` denotes the positional embeddings, $Q$ and $K$ are learnable linear projections. In our implementations, we keep low-dimensional projections $Q : \mathbb{R}^1 \mapsto \mathbb{R}^{16}$ and $K : \mathbb{R}^1 \mapsto \mathbb{R}^{16}$.

Compared to the linear classifier, the bi-linear and attention-based classifiers can capture relationships between shapelet concepts while retaining some level of interpretability. For example, the bi-linear classifier could express "shapelet $s_k$ and $s_j$ both exist in $x_i$". Although the attention mechanism (i.e., $Q$ and $K$ projections) may not be considered as fully interpretable, `attention`$(Q(\mathbf{p}_i) +$ `PE`$, K(\mathbf{p}_i) +$ `PE`$, p_{i,k})$ produces predicates that combines shapelet predicates based on attention weights, which preserves some level of interpretability

In Table 16 we compare the three SBM classifier architectures on a subset of the UEA datasets. Our results indicate that the advanced classifiers improve performance for both SBM and InterpGN on certain datasets. Considering the simplicity, good interpretability of linear classifier and its comparable performance to more advanced classifiers on most datasets, we suggest that it is still favorable to use linear classifier in practice; however it is good to try more advanced classifiers to achieve better performance. We will also leave the exploration of other advanced classifiers (such as energy-based or graph-based ones) for future work.

Table 16: Classification accuracy for the three SBM classifier architectures. The best accuracy for each dataset and model is marked with bold. The PEMS-SF dataset is omitted since due to the quadratic memory cost in bi-linear and attention-based classifiers and its high dimensionality, which fail to run.

| Model: | SBM | | | InterpGN | | |
|---|---|---|---|---|---|---|
| Classifier: | Linear | Bi-linear | Attention | Linear | Bi-linear | Attention |
| EthanolConcentration | 0.301 | 0.299 | **0.302** | 0.298 | **0.307** | 0.290 |
| FaceDetection | **0.657** | 0.638 | 0.644 | **0.659** | 0.642 | 0.642 |
| Handwriting | **0.357** | 0.314 | 0.282 | **0.617** | 0.605 | 0.609 |
| Heartbeat | 0.736 | **0.738** | 0.733 | **0.779** | 0.736 | 0.745 |
| JapaneseVowels | **0.963** | 0.951 | 0.936 | **0.990** | 0.898 | 0.989 |
| SelfRegulationSCP1 | 0.864 | 0.859 | **0.866** | **0.917** | 0.883 | 0.914 |
| SelfRegulationSCP2 | 0.522 | 0.530 | **0.566** | 0.578 | 0.562 | **0.582** |
| SpokenArabicDigits | **0.994** | 0.990 | 0.987 | **0.997** | 0.989 | 0.995 |
| UWaveGestureLibrary | **0.913** | 0.873 | 0.854 | **0.911** | 0.902 | 0.902 |

## C.4 TIME SERIES EXTRINSIC REGRESSION

To further investigate the generalizability and limitations of the proposed methods, we apply them to time series extrinsic regression tasks using datasets from the Monash, UEA, and UCR Time Series Extrinsic Regression Repository (Tan et al., 2021). Because the proposed methods are inherently classification models, we recast the regression tasks as ordinal regression problems. Moreover, given that some datasets exhibit highly imbalanced target value distributions, we discretize the target variable into 10 equally sized bins spanning the minimum and maximum target values of the training split for each dataset.

Since the models produce probabilistic predictions over the ordinal categories (i.e., the bins), we employ the Continuous Ranked Probability Score (CRPS) (Gneiting & Raftery, 2007), which is also widely used in probabilistic forecasting (Aksu et al., 2024), as the performance metric for training and evaluation. Formally, the CRPS is defined as:

$$\mathrm{CRPS}(\hat{\mathbf{r}}, b) = \sum_{i=1}^{B} \left( \sum_{j=1}^{i} \hat{r}_j - \mathbb{1}(j \geq B) \right)^2, \tag{14}$$

where $\hat{\mathbf{r}}$ is the predicted probabilities for each bins, $b$ is the true target bin, and $B$ is the total number of bins. Table 17 summarizes the results obtained for the SBM, InterpGN, and three deep learning models. It should be noted, however, that this experimental setup is preliminary and may require further refinement; our primary objective is to demonstrate the feasibility of extending our proposed methods to regression tasks.

Table 17: Continuous Ranked Probability Score (CRPS) of the methods on the time series extrinsic regression datasets. The best result for each dataset is marked with bold, and the second best is marked with italic.

| Model: | SBM | InterpGN | | | Deep Learning | | |
|---|---|---|---|---|---|---|---|
| DNN Type: | - | FCN | PatchTST | TimesNet | FCN | PatchTST | TimesNet |
| AppliancesEnergy | *0.844* | **0.834** | 1.274 | 0.937 | 0.878 | 1.331 | 1.135 |
| BIDMC32HR | **0.275** | 0.550 | *0.369* | 0.378 | 0.981 | 2.026 | 1.505 |
| BeijingPM10Quality | 0.549 | 0.425 | 0.835 | 0.441 | **0.415** | 1.157 | *0.424* |
| BeijingPM25Quality | 0.477 | 0.332 | 0.569 | 0.332 | **0.313** | 1.440 | *0.317* |
| BenzeneConcentration | **0.966** | 1.038 | 2.997 | 1.250 | *1.031* | 2.234 | 1.258 |
| Covid3Month | 0.013 | 0.014 | 0.017 | 0.017 | **0.000** | **0.000** | **0.000** |
| FloodModeling1 | 0.006 | 0.006 | 0.008 | 0.008 | **0.000** | **0.000** | **0.000** |
| HouseholdPowerConsumption1 | 0.456 | **0.405** | 1.318 | 0.935 | *0.409* | 1.079 | 0.807 |
| IEEEPPG | **1.122** | 2.285 | 2.896 | 2.862 | *2.250* | 2.619 | 3.077 |
| LiveFuelMoistureContent | **0.491** | 0.501 | 0.649 | 0.648 | *0.494* | 0.644 | 0.644 |
| NewsHeadlineSentiment | 0.001 | **0.000** | **0.000** | **0.000** | **0.000** | **0.000** | **0.000** |

