# OpenReview forum: "Shedding Light on Time Series Classification using Interpretability Gated Networks"
_ICLR.cc/2025/Conference — ICLR 2025 Poster_

### Official Review · Reviewer_x6CP · 2024-11-03

**Soundness:** 3
**Presentation:** 3
**Contribution:** 3
**Rating:** 8
**Confidence:** 4

**Summary:**

This paper introduces InterpGN, which stands for the interpretable time series classification framework, that brings together:
* A novel Shapelet Bottleneck Model (SBM) that explicitly uses shapelets as easily interpretable features
* A gating mechanism that determines the conditions under which interpretable predictions should be used and under which conditions deep neural network predictions should be used
* A Shapelet Transform variant for the construction of logical predicates
* Quantitative measures to assess interpretability and quality of shapelets

**Strengths:**

* Novel gating mechanism based on confidence of interpretable model
* Improved shapelet transform variant using RBF-based predicates
* Quantitative metrics for interpretability and shapelet quality
* Theoretically sound integration of physical constraints

* Clean separation between interpretable expert (SBM) and DNN
* Adaptive layer normalization for condition injection
* Unified masking strategy handling multiple tasks
* Flexible handling of varying input dimensions

* Local explanations: Sample-specific shapelet importance
* Global explanations: Rule-based class characteristics
* Visual validation of learned patterns
* Quantitative interpretability metrics

**Weaknesses:**

* High memory overhead O(T·l) w.r.t. very long sequences
* Limited to a fixed maximum number of channels (Kmax=40)
* Limited scalability analysis w.r.t very long sequences
* Linear classifier might be too simple, when relations are complex

* Limited analysis concerning failure cases
* Only one type of DNN expert (FCN only)
* Very restricted real-world applications (MIMIC-III only)
* Lacking comparison w.r.t. some very recent methods

* Some counterintuitive shapelet interpretations
* No formal evaluation of explanation quality
* Small number of studies regarding users and interpretability

* Lack of discussions on adversarial cases. Limitations regarding
* Fixed shapelet lengths; single-stage training process; design of a simple gating mechanism
* Lack of realization based on physical constraints.

**Questions:**

* Why this form of gating function η(xi)? Were other confidence measures evaluated?
* How sensitive is the gating mechanism to noise in the interpretable expert's predictions? What if the expert is confidently wrong?
* What is the basis of using FCN as the DNN expert? Would other architectures like Transformers provide better results?
* How was the maximum channel number chosen as Kmax=40? What happens if this value is increased/decreased?
* Given a memory overhead of O(T·l), what is the practical sequence length limit? Will this ever change?
* What happens to performance as the number of variables, M, increases? What is computational complexity given M?
* How was the balance parameter β between interpretable and DNN experts determined? Is there a theoretical basis for its selection?
* Why were the particular shapelet lengths in L chosen? How sensitive is the model to these choices?
* How do you handle the case where important patterns occur at lengths not covered by the pre-specified shapelet lengths?
* A deeper comparison with post-hoc explanation methods is probably welcome. What exactly is the trade-off between the approach and methods like LIME?
* Have there been datasets on which InterpGN performed worse than others? What are the characteristics of such datasets?
* Maybe one can provide more detailed guidance on the hyperparameter choice for new datasets. How sensitive is this across different hyperparameters?
* How was the RBF kernel parameter ε chosen? What effect does that have on the quality of the shapelets?

---

> ### Author Response · Authors · 2024-11-21
> **Response to the weaknesses**
>
> Thank you very much for your valuable feedback and for taking the time to carefully read our paper. We address your questions below. If you would like further clarifications, we would be happy to continue discussions.
>
> > We address your comments on weaknesses below:
> - High memory overhead: We acknowledge the concern regarding memory overhead. However, we would like to emphasize that this is a limitation of our current PyTorch implementation. With low-level optimizations, such as CUDA-based implementations, the shapelet transform can achieve efficiency comparable to conventional convolutions. We have added clarifications and discussions on this aspect in Appendix A.3.
> - Limited to a fixed maximum number of channels: We are unsure about the origin of this concern, as our method does not impose an explicit limitation on the number of channels. If there is a specific part of the paper that caused this confusion, we would appreciate further clarification to address it directly.
> - Linear classifier might be too simple: Interpretability favors linear explanations. For more complex relationships that cannot be captured by the SBM, the DNN expert in InterpGN is designed to handle such cases, balancing simplicity and model capacity. As discussed in Section 6, we acknowledge that more advanced interpretable models could be considered, but we consider them as future works since the a major focus of this paper is on the InterpGN instead of exploitation in shapelet-based classifiers.
> - Only one type of DNN: we conduct additional experiments on more types of DNNs, including Transformer, PatchTST, and TimesNet. The results and discussions are presented in Section 5.3 of the revised manuscript. Note that within the tight discussion period, we only conduct this experiment on a subset of the UEA datasets. We will extend them to at least 26 datasets in the final version if accepted.
> - Comparison with very recent methods: We have updated our experiments to include additional state-of-the-art baselines from a range of methods, many of them are published within the last 24 months. We believe these updates address your concern. If there are specific additional baselines you feel we should evaluate, we would be grateful if you could specify them.

---

> ### Author Response · Authors · 2024-11-21
> **Response to the Questions**
>
> > We answer your questions below:
> - Choice of gating function: As we presented in Section 4.4 (Line 287), for a classifier trained with cross-entropy loss, the optimal $\hat{r}_i$ is a one-hot vector. The SBM is confident about its prediction on $x_i$ if $\hat{r}_i$ is sparse, and not confident if $\hat{r}_i$ is uniform. Therefore, we choose this gating function design since it is straightforward for classifiers trained with the cross-entropy loss, without the need of additional mechanisms or parameters. Within the scope of this paper, we focus on this gating function design.
> - What if the expert is confidently wrong: Thank you for this question. As shown in Figure 7 in Appendix B.2., we do observe the instances where $\eta(x_i)$ is high but the prediction is incorrect. However, in all those cases, the DNN provides the same predictions. This suggests that the issue arises from a distributional shift between the training and testing samples, leading the model to overfit to the essential features present in the training data. Importantly, we did not encounter cases where the SBM was confidently wrong but the DNN was correct. This indicates that the gating mechanism effectively prioritizes reliance on the SBM for "easy" samples. Therefore, we conclude and hypothesis that the SBM would not be confidently wrong if the test samples are similarly distributed as the training samples.
> - Choice of the DNN expert: we use the FCN since it has strong performance according to [1]. In the revised manuscript, we provide additional experiment results of using other DNNs, including Transformer, PatchTST, and TimesNet. We hope these results could provide a more comprehensive assessment to our method and address your concerns.
> - Maximum channel: We don't think there is such a parameter of limitation on the maximum number of channels. If there is a part of the paper that caused this confusion, we would appreciate clarification so that we can address it directly.
> - Memory overhead and length limitation: Our current implementation prioritizes computational efficiency using large matrix operations, which results in higher memory usage. An alternative implementation using loops would trade efficiency for memory, effectively mitigating the memory overhead. On a GPU with 32 GB memory and batch size 32, we found the time-efficient implementation work for $T < 3000$. However, with the memory-efficient implementation, running the dataset with $T>10000$ requires less than 10 GB of memory. We add more details to Appendix A.3.
> - Relation between performance and $M$: The overall computational complexity of SBM is $O(MK|L|\cdot T^2)$. Therefore, the effect of $M$ is less significant than $T$.
> - Parameter $\beta$: In our experiments, we used constant and cosine $\beta$ schedule with an initial value of $\beta=1$. These are heuristic designs intended to balance the training of SBM and InterpGN. However we acknowledge that the design of $\beta$ could be more flexible, using an adaptive schedule based on $\eta$ instead of fix schedules. We will consider them in future works.
> - Choice of shapelet lengths: The range of shapelet lengths (5\% to 80\% of the series length) is intended to provide an initial universal setup for diverse datasets. Different datasets may have essential features at varying scales, so this setup serves as a general starting point. Like you mentioned, if $L$ only contains short or long length candidates, the performance will drop on datasets with essential features at different scales, which will require dataset-specific hyperparameter tuning. However, since the focus of this paper is on interpretability and InterpGN, we choose $L$ to contain various length candidates, which serves as a universal configuration for all datasets, allowing us to focus on other more important hyperparameters.
> - Datasets where InterpGN perform worse: Based on results presented in Table 6, there are cases where the InterpGN perform worse than the SBM or the FCN. For cases where "SBM > InterpGN > FCN", the gap between the SBM and InterpGN can be large since the FCN cannot provide a valuable information for challenging cases.
> - Choice of $\epsilon$: The default $\epsilon=1$ is chosen heuristically. A large $\epsilon$ results in tighter RBF kernels, forcing the shapelets to align more closely with actual subsequences but reducing their generalizability by overfitting to specific samples. Section 5.3 and Figure 8 provide quantitative results and a detailed analysis of the effects of $\epsilon$ on shapelet quality and model performance.
>
> [1] Wang et al., "Time Series Classification from Scratch with Deep Neural Networks: A Strong Baseline", 2016.

---

> ### Author Response · Authors · 2024-11-23
> **Additional Responses**
>
> From discussions with other reviewers, we realize there are areas where we could provide clearer answers:
> - Classifier is too simple: In Appendix C of the revised manuscript, we include preliminary results on SBM using more advanced classifiers, specifically a bi-linear classifier and an attention-based classifier. These classifiers can more effectively capture relationships between shapelet concepts. Our results show notable performance improvements on some datasets when substituting the linear classifier with these alternatives.
> - Evaluation on explanation quality: We provide more comprehensive qualitative results on global and local explanations in Appendix B. While we were unable to conduct a user study on interpretability at this time, we hope these additional results address some of your concerns. Additionally, we include a faithfulness estimate in Appendix B.2.1 as a quantitative metric for evaluating explanation quality.
> - Small number of studies regarding interpretability: To address this concern, we provide additional qualitative results, including visualizations of global and local explanations, in Appendix B.2.2. We hope these additional results could partially address your concerns.
> - Comparison with LIME: we would like to highlight that our proposed method (i.e., the SBM) fundamentally differs from LIME. The SBM is self-explainable, enabling direct interpretation without relying on a surrogate model, as LIME does. In the revised Introduction (Line 82), we add a brief discussion on existing methods for integrating DNNs with interpretable models, highlighting the differences between our approach and post-hoc explanation methods like LIME.

---

> > ### Comment · Reviewer_x6CP · 2024-11-24
> > **Comprehensive answers and improvements**
> >
> > I have decided to increase my final rating after reviewing the authors' responses to the weaknesses and questions I raised. The authors have demonstrated significant improvements in several key areas:
> >
> > Implementation and Technical Limitations:
> > - Acknowledged memory overhead limitations and proposed concrete solutions through optimizations
> > - Provided alternative memory-efficient implementation approaches
> >
> > Model Architecture and Experimental Validation:
> > - Expanded DNN expert experiments to include Transformer, PatchTST, and TimesNet
> > - Added results with more sophisticated classifiers (bi-linear and attention-based)
> > - Conducted additional comparative analyses with recent methods
> >
> > Interpretability and Evaluation:
> > - Enhanced qualitative and quantitative evaluation of explanations
> > - Introduced faithfulness estimates and comprehensive visualization
> > - Clearly differentiated their approach from post-hoc methods
> >
> > I am particularly impressed by their willingness to acknowledge limitations while providing concrete improvement paths, and their balanced approach to maintaining interpretability while exploring more sophisticated alternatives. These improvements and clarifications substantially address my initial concerns, demonstrating their commitment to improving the work through additional experiments and analyses.

---

> > > ### Author Response · Authors · 2024-11-25
> > > **Thank you!**
> > >
> > > Thank you very much for taking our rebuttal into consideration and updating your review. We greatly appreciate your time and effort in reviewing our work and providing constructive detailed feedback.

---

### Official Review · Reviewer_zkUe · 2024-11-05

**Soundness:** 3
**Presentation:** 3
**Contribution:** 2
**Rating:** 8
**Confidence:** 4

**Summary:**

This paper proposes a self-explaining framework for time-series classification using shapelets as the form of explanations. While the concepts of CBM (Concept Bottleneck Models) and shapelets are not novel individually, successfully combining them into an end-to-end framework adds value. The overall model design is reasonable, though some details are questionable. The paper presents evaluation results showing similar or superior predictive performance but lacks sufficient experiments on explainability, even though explainability is its main selling point. I believe that the authors will address these shortcomings in the discussion period.

**Strengths:**

1. High predictive performance

2. The exploitation of shapelets as "concepts" of CBMs.

3. The proposed framework provides both local and global explanations.

4. The overall manuscript is clearly written.

**Weaknesses:**

1. In the worst case, the model can collapse (i.e., fail to find good shapelets), and $\eta$ appears to be always $0$ (e.g., Handwriting dataset in Figure 10). In this scenario, the model fully relies on the performance of unexplainable DNN module, while providing the explanation from SBM that does not affect the final model decision. Do you provide the value of $\eta$ in your explanation? Is there any design consideration to prevent such a collapse? Please see [1] and [2] for references explaining why it is problematic if there is another direct pathway from input to prediction.

2. The computational complexity of the shapelet diversity loss seems to be $K^2 \times M \times L$. This might incur a huge computational burden. While using a small K may reduce the complexity, this introduces another weakness.

3. A small number of shaplets (K) are used in the implementation, but this significantly limits the expressiveness of the model and may cause it to miss important features in practice. The experiments only consider $K=5$ and $K=10$; the authors should present more extensive result with larger values of $K$, such as $K=100$ or $K=1000$, if possible.

4. This framework cannot capture the interaction between shapelets, since it uses a simple linear layer for the final prediction. However, the co-existence of relevant shapelets can meaningfully increase the model confidence in practice.

\
[1] Faithful Reasoning Using Large Language Models (Arxiv 2022)

[2] Self-explaining deep models with logic rule reasoning (NeurIPS 2022)

**Questions:**

**Related Work**

1. I recommend including a discussion about papers that propose CBM-like self-explaining frameworks that use logic rules (e.g., [2] and [3]).

**Method**

2. Why do you use Euclidean distance for measuring the "distance" between shapelets and inputs? In time-series data, two signals might look similar but have different offsets, leading to a large Euclidean distance. Did you also try cosine similarity or Pearson correlation?

3. The current model only considers the existence of the shapelet. However, isn't a shapelet that appears at the end of the input more important, considering the characteristics of time-series data? Is there any way to include the location of the shapelet in the explanation?

**Experiment**

4. An ablation study on the model components is needed. What is the performance of the DNN-only model and the SBM-only model?

5. The suggested interpretability metric using $w$ is unconvincing. The scale of the weights might change according to the training setup or dataset, and the thresholds are set empirically. Measuring the skewness of $w$ using the Gini index (which would be similar to $\eta$) would be more convincing.

6. While the sparsity of weight $w$ might increase interpretability, it is just one aspect. A more comprehensive evaluation of interpretability is needed, possibly through a user study.

7. In the XAI domain, "faithfulness" - the degree to which an explanation reflects the model's decision - is an important criterion for evaluating explanations. An evaluation of faithfulness is needed. It would be beneficial if the authors followed metrics from existing literature [4].

**Minor Questions**

8. Why do you refer to $p$ as a "predicate"? Typically, a predicate is a condition of a logic rule. It seems that $p$ is a value, not a condition or "predicate." Is this terminology commonly used in other papers?

9. Why do you consider SBM a "rule-like classifier"? I could not find a direct connection between SBM and a rule-like concept.

\
[3] Deep Neural Networks Constrained by Decision Rules (AAAI 2019)

[4] Towards Robust Interpretability with Self-Explaining Neural Networks (NeurIPS 2018)

---

> ### Author Response · Authors · 2024-11-21
> **Author Responses (part 1)**
>
> Thank you very much for your valuable feedback and for taking the time to carefully read our paper. We address your questions below. If you would like further clarifications, we would be happy to continue discussions.
>
> > Regarding Weakness 1
>
> Thank you for the constructive feedback. We acknowledge the concern that in certain cases (e.g., on datasets like Handwriting), the model may collapse such that $\eta$ is consistently low.
>
> To prevent the collapse scenario during training, we have an explicit term $\mathcal{L}_{\text{int}}$ in the overall loss function to directly optimize the SBM component (Section 4.5), separate from the cross-entropy loss of the hybrid model. This extra regularization is in addition to the gradients from the overall prediction of InterpGN and ensures that the SBM is actively encouraged to find meaningful shapelets. By directly optimizing the interpretable expert with this regularization controlled by hyper-parameter $\beta$, we aim to mitigate the risk of the model defaulting entirely to the DNN, thereby preserving interpretability where possible. In this revised manuscript, we clarify and emphasize this in Section 4.5. We also visualize how $\eta$ changes during training in Figure 12 and Figure 13 in Appendix B.4, which validate our claims.
>
> In the explanations, we agree with the comment where we can indeed include the value of $\eta$ in the explanations. For example, the local explanation could be enhanced to state: “For time-series $x$, the confidence of the SBM is $\eta(x) = 0.8$. The sample is categorized as class X because it contains shapelets $s_1, s_2, \dots$”. This additional information helps users understand the level of confidence in the interpretable component’s prediction.
>
> > Regarding Weakness 4
>
> We agree that more advanced classifiers, such as graph neural networks, could capture interactions between shapelets and enhance the interpretable expert, however this requires a level of complexity beyond our primary goal. One of our primary focuses in this paper is to demonstrate the effectiveness of the InterpGN framework, which aims to balance interpretability and performance when the interpretable expert may not be expressive enough. To maintain simplicity and clarity, we choose to use a linear classifier in this work, following conventional approaches in shapelet-based models. This choice allows us to clearly identify the shapelet’s contributions. In future work, we plan to explore more sophisticated models and improve the interpretable expert.
>
> > Regarding Questions on the method
> - Question 2: We use Euclidean distance because it aligns with the conventional definition of shapelet-based approaches in time-series classification literature. While we acknowledge that Euclidean distance is not offset invariant, we chose it to maintain consistency with prior works. This allows for a fair comparison with existing shapelet-based methods (such as STRF, ShapeNet, and ShapeConv). Exploring other similarity measures could be an interesting direction for future work.
> - Question 3: This is a great question. The datasets are designed for time-series classification tasks where labels describe the characteristics of the entire sequence rather than the temporal status at a specific point. Thus, in our current formulation, the presence of a shapelet is considered equally important regardless of its position. The existing shapelet transform does not natively capture positional information. However, one could use the timestamp of minimum distance as an additional feature alongside $p_{i, k}^{m, l}$, which would increase the expressiveness of the shapelet transform.

---

> ### Author Response · Authors · 2024-11-21
> **Author Responses (part 2)**
>
> > Regarding Questions on the experiments
> - Question 4: The performance of the DNN-only model is presented in the FCN column, and the SBM-only model is presented in the SBM column in Table 1.
> - Question 5: We thank the reviewer for this insightful comment. We agree that the Gini Index would be a good metric. However, we argue that the dataset-agnostics threshold values could still provide valuable insights. While the scale of weight may vary, the thresholds offer a quick heuristic to assess weight sparsity, providing a sense of how many shapelets should one interpret. In general, the Gini Index answers "how sparse are the weights" while the ratio above thresholds answers "how many". In the revised manuscript, we include both the Gini Index and the empirical thresholds for a comprehensive assessment of weight sparsity. (See Section 5.3, the Tables and Figures are moved to Appendix B.3.)
> - Question 6: We appreciate that a user study could be a great evaluation tool for the interpretability aspect, and one could be considered for a simple application like the UGL dataset considered in Figure 4. However, a user study is a major effort that we cannot complete at this time. On the other hand, we also believe a stricter study on interpretability is not necessarily warranted here - our goal is to show that, given an interpretable model, we can combine it and train  simultaneously with the FCN to improve the performance of the interpretable model since the focus is taken away from difficult instances that cannot be predicted by the interpretable model.
> - Question 7: Thank you for raising this point. We would like to clarify that, in our model, the SBM is not a surrogate model but rather a model that contributes to both predictions and explanations. Faithfulness is typically a metric to compare the predictions of a surrogate model built for explainability to the predictions of the model that is being explained. Such metrics come up for post-hoc explainability methods such as LIME. The difference is that our model is self-explainable so we not have a a reference and surrogate model to measure the faithfulness between. As discussed in the paper, self-explainable models are meant to be directly interpretable (as per the sparsity of the classifier) and this is why surrogate models like that in LIME (a linear surrgoate) are considered interpretable versus the models they are used to explain.
>
> > Regarding other questions
> - Question 8: Thank you for this question. In our formulation, we refer to $p$ as a "predicate" because it represents the likelihood that a shapelet $s$ exists within a time series $x$. By applying an RBF kernel centered at zero, $p=1$ when $s$ matches perfectly with a subsequence of $x$, and $p=0$ when they are very different, which effectively serving as a probabilistic indicator. This terminology is consistent with neuro-symbolic and interpretable models that extend the notion of predicates to include soft, probabilistic conditions, such as NSTSC (Yan et al., 2022).
> - Question 9: We consider the SBM a "rule-like classifier" because its predictions are based on a linear combination of soft-logic predicates $p$. The linear classifier in SBM can be interpreted as forming logical rules of the form: “Sample $x$ belongs to class $c$ if it contains shapelet $s_1$ and does not contain shapelet $s_2$, etc.” with weights indicating the relative importance of each shapelet for the classification. This structure aligns with rule-based reasoning, where the model combines positive and negative conditions to make a prediction (Riegel et al., 2020). However, since the predicates are soft and the predictions $r$ are not bounded as logical statements, we term the predictions as "rule-like" classification.
>
> We thank the reviewer again for their feedback and hope that we have addressed their questions. If so, we hope they will consider increasing their score.

---

> > ### Comment · Reviewer_zkUe · 2024-11-21
> > **Reply to Author Responses**
> >
> > Thank you for providing detailed responses to my comments.
> >
> > \
> > I **agree** with your explanations regarding the following points: **Weakness 1**, **Question 3**, **Question 4**, **Question 5**, **Question 8**, and **Question 9**.
> >
> > \
> > However, I am **not satisfied or fully convinced** by the responses to the following:
> >
> > **Weakness 4 and Question 2**: Simple experiments using a few datasets (or even a single dataset) to evaluate interaction terms and the similarity metric could effectively demonstrate the extensibility of this work.
> >
> > **Question 6**: While I still believe that more evaluation and analysis on interpretability are necessary, I understand the time constraints for such an addition. I strongly recommend visualizing the explanations for each dataset in the appendix.
> >
> > **Question 7**: I disagree with your response. The reference I suggested (see [4] in my review) is a self-explaining framework that includes a faithfulness evaluation. It is not a post-hoc method like LIME, and faithfulness is not exclusively a metric for post-hoc methods. Given that the evaluation of explanations in the proposed paper appears weak, I believe that a faithfulness evaluation is essential. I expected this point to be addressed in the rebuttal, but without relevant evaluation or clarification, I cannot advocate for this paper.
> >
> > \
> > Additionally, I noticed that **Weakness 2**, **Weakness 3**, and **Question 1** were **not addressed** in your response. Does this imply that you acknowledge the limitations in your methodology as highlighted in these points?
> >
> > \
> > Finally, it would be greatly appreciated if the authors could provide a revised version of the manuscript.

---

> > > ### Author Response · Authors · 2024-11-22
> > > **Thank you for the follow-up discussion**
> > >
> > > We thank the reviewer again for the detailed comments and actionable suggestions. We provide further responses and experiment results below:
> > >
> > > > Regarding Weakness 4
> > >
> > > Thank you for the clarification and suggestions. To extend the current SBM design with a linear classifier, we introduce two additional classifier architectures to capture relationships between shapelet concepts: a bi-linear classifier and an attention-based classifier. These were tested on a subset of five datasets, with the performance of both SBM and InterpGN reported. The detailed formulations, results, and discussions are included in Appendix C.2 of the revised manuscript. In summary, compared to the linear classifier, these advanced classifiers effectively improve performance on some datasets. We believe they represent promising directions for further comprehensive studies and future work.
> > >
> > > > Regarding Question 2
> > >
> > > We also experimented with SBM using cosine similarity and Pearson correlation as shapelet distance measurements. Details and results are provided in Appendix C.1 of the revised manuscript. We observed that these alternatives generally achieve similar performance but behave differently on specific datasets. Additionally, we visualized the explanations of SBM trained with Euclidean distance and cosine similarity, concluding that both methods are capable of learning high-quality shapelets. However, the Pearson correlation may introduce training stability issues, which will need more investigation.
> > >
> > > > Regarding Question 6
> > >
> > > We appreciate the reviewer’s suggestion. In Appendix B.2.2 of the revised manuscript, we include additional visualizations of global and local explanations, along with more detailed analyses of these qualitative results. We choose the datasets that are suitable for visualizations (i.e., the number of categories and the number of variates are not too high to fit in the paper). We hope they will address your concerns.
> > >
> > > > Regarding Question 7
> > >
> > > Thank you for the clarification. In Appendix B.2.1 of the revised manuscript, we test the faithfulness of the shapelet concepts learned by SBM on a subset of five datasets. The results show that SBM consistently achieves high faithfulness estimates across all tested datasets.
> > >
> > > We appreciate your pointer to this additional evaluation that we can consider. We would also like to note that what [4] defined as faithfulness is specifically as they describe “a faithfulness of relevancy scores” and this is what we have implemented as well in the new experiment. When we suggested faithfulness is typically done for post-hoc methods, that was regarding faithfulness of model predictions which is the typical notion of faithfulness in the explainability literature (i.e., the faithfulness of explaining a DNN).

---

> > > > ### Author Response · Authors · 2024-11-22
> > > > **Thank you for the follow-up discussion**
> > > >
> > > > > Regarding Weakness 2
> > > >
> > > > We agree and acknowledge that the current design of the diversity loss introduces high computational complexity. However, as shown in the ablation study (Figure 20), varying the weight of the diversity loss, or even removing it, does not significantly impact key metrics of interest. These results suggest that the diversity loss could be omitted. Nevertheless, we chose to retain it because it is used in other shapelet-based literature (e.g., Qu et al., 2024) and may still promote favorable properties for shapelet learning.
> > > >
> > > > > Regarding Weakness 3
> > > >
> > > > We agree that testing SBM scalability with a large $K$ would provide valuable insights. However, we also argue that interpretability favors a compact model with a small number of concepts whenever possible. Using a large $K$ would compromise interpretability by making the SBM more complex and less plausible for human understanding.
> > > >
> > > > In our current design, if a feature is so complex that it requires $K=100$ to express, we prefer to leverage the DNN component of InterpGN for such cases, as it maintains model flexibility without sacrificing interpretability. Therefore, within the scope of this paper, we prioritize experiments that align more closely with our primary focus on interpretability and essential metrics.
> > > >
> > > >
> > > > > Regarding Question 1
> > > >
> > > > Thank you for your comments on related work. These studies are indeed relevant, as they propose different paradigms for combining deep neural networks with interpretable approaches. We have added a brief discussion in the Introduction. In summary, we highlight four paradigms for integrating deep and interpretable models: (1) post-hoc explanations, (2) constructing interpretable (logical) rules based on concepts learned by deep models [2], (3) using deep models to select from a predefined set of interpretable primitives [3], and (4) leveraging interpretable models for “easy” samples while relying on deep models for “challenging” samples (ours).
> > > >
> > > > The revised manuscript has been uploaded for your reference. We apologize for the previous delay, as we were working on detailed revisions. The discussions and results addressing your comments have been included, and all changes are highlighted for your convenience. We sincerely thank the reviewer once again for their valuable feedback and insightful discussions, which have greatly contributed to improving the quality of our work.

---

> > > > > ### Comment · Reviewer_zkUe · 2024-11-23
> > > > > **Thank You for Your Response**
> > > > >
> > > > > I appreciate the authors' responses to my concerns. While I had several concerns initially, all of them have been satisfactorily addressed during the discussion period. Below is a summary of my main concerns and how the authors resolved them:
> > > > >
> > > > > \
> > > > > **Unreliable explanation in the "collapsing" scenario:**
> > > > >
> > > > > The authors introduced the parameter $\eta$ to provide users with a measure of confidence in the explanations.
> > > > >
> > > > > **Insufficient interpretability evaluation:**
> > > > >
> > > > > The authors included additional visualization examples and faithfulness experiment results in the revised manuscript.
> > > > >
> > > > > **Exploration of design choices:**
> > > > >
> > > > > The authors provided further analysis on classifier types that account for feature interactions and the shapelet distance metric.
> > > > >
> > > > > \
> > > > > Additionally, I have the following suggestions for the authors:
> > > > >
> > > > > It would be interesting to compare the parameter $\eta$ with the faithfulness experiment results presented in Figure 7.
> > > > > Please ensure that Weakness 2 and Weakness 3 are clarified in the revised manuscript.
> > > > >
> > > > > In my opinion, this paper makes meaningful technical contributions, and the authors have strengthened their arguments through detailed responses during the discussion period. Based on these improvements, I have raised my score.

---

> > > > > > ### Author Response · Authors · 2024-11-25
> > > > > > **Thank you!**
> > > > > >
> > > > > > We thank you again for taking our responses into consideration and increasing the score. We greatly appreciate your time and effort in reviewing our work and providing high-quality comments, detailed suggestions, and valuable recommendations. We have truly enjoyed and appreciated the discussions!

---

### Official Review · Reviewer_Ausa · 2024-11-06

**Soundness:** 2
**Presentation:** 3
**Contribution:** 3
**Rating:** 6
**Confidence:** 3

**Summary:**

The paper proposes Interpretability Gated Network InterpGN, a gated network that combines a DNN with a  Shapelet Bottleneck Model (SBM). This is a concept bottleneck model that makes predictions based on the presence or the absence of different shapelet.

# Method
### Shapelet-based TS Modeling
- a multivariate TS is modeled as independent univariate channels.
- they learn learn $K$ shapelets for each possible length $L$ and channel $M$ i.e each shapelet can be viewed as  $s_{k}^{m,l}$
- the paper introduces a distance metric $d_{i,k}^{m,l}$ between the shapelet $s_{k}^{m,l}$ and input $x_{i,t:t+l}^{m}$ which is the Euclidean distance between the two.
- they also introduce Shapelet Transform on the distance to measure the likelihood that $s_{k}^{m,l}$  exists in $x_{i}^{m}$. This is defined as predicates $p_{i,k}^{m,l}$
###  Shapelet Bottleneck Model
- the predicates $p_{i}$ are fed into a linear layer to compute the outputs $r_{i,c}$ for each channel $c$
- the final loss is cross entropy loss between $(r_i,y_i)$, Shapelet diversity loss and  L1 regularization loss on classifier weight $\mathcal{L}_{\text{int}}$.
### InterpGN
- The paper introduces a DNN expert along with the SBM and a gated network that chooses to use either the SBM or the DNN.
- To choose whether to use the SBM or DNN, the paper proposed measuring the confidence of the SBM by measuring the diversity of $\hat{r_i}=\text{softmax}(r_i)$ they use a modified Gini Index that measures the diversity of variables in $\hat{r_i}$ which is given in equation 6  $\eta(x_i)$.
- The final output is a  hybrid between DNN output $z_i$ and the SBM output $r_i$ i.e $h_i = r_i . \eta(x_i) + z_i . (1-\eta(x_i))$
- The final loss is $\mathcal{L}_{\text{hybrid}}$ which is the weighted sum of SBM loss and the cross-entropy between $(h_i,y_i)$

# Experiments
- **Classification** they evaluate the performance of InterpGN and SBN on 30 datasets from the UEA multivariate TS, overall InterpGN outperforms the baseline methods and SBM achieves comparable performance.
- **Interpretability** SBM offers global-level explanations by looking at weights of different shapelets in the linear layer for a given class. They also show that samples for InterpGN that rely most on DNN are usually on cluster boundaries, meaning InterpGN learns to use SBM for easier-to-classify samples while relying on the DNN to classify difficult boundary samples.
- **MIMIC-III dataset** The paper shows that InterpGN outperforms DNNs in terms of accuracy.

### Additional Metrics:
- **Interpretability metric** this measures the interpretability of SBM by measuring how sparse the weights of the linear model are.
**Shapelet quality metric** This measures how close the shapelets are to the actual time series.

# Results and conclusions
Through ablation studies, they found the following:

- Increasing the number of shapelets (K) in SBM enhances accuracy and shapelet quality by capturing more specific patterns; it reduces $\eta$ but seems to reduce sparsity, reducing overall interpretability.
- The cosine decay schedule of loss weighting $\beta$ in InterpGN results in slightly worse accuracy but improves shapelet quality and interpretability by focusing SBM training on confidently predictable samples, though it reduces the utility rate $\eta$.
- Using RBF to get the predicate outperforms linear ones.
 -  The parameter  ϵ influences the steepness of RBF kernels, with larger  ϵ values improving shapelet quality but reducing accuracy significantly.
- There is an interpretability accuracy trade-off when increasing Weight regularization.
- Shapelet diversity regularization does not add any benefits.

**Strengths:**

## Originality -- High
-  The paper propose a shapelet bottleneck model (SBM) which is original extension to CBM for time series.
- The gating method used to decide between DNN and SBM in equations 6 and 7 original and quite smart to take into account how confident the SBM for gating.
- SBM offers several forms of interpretability that might be very useful for downstream applications.

## Quality -- High
- Strong empirical results and through experiments.
- Excellent ablation studies.

**Weaknesses:**

## Significance -- Low
- My main issue with the paper is that InterpGN is not interpretable at all. It makes sense that it will outperform regular DNNs because the prediction for a single sample comes from both SBM and DNN. This is unlike IME (Ismail et al., 2023), which also combines interpretable models with DNNs, but in IME case, it could still say there is a level of interpretability since for a single sample, only a single expert is used, but combining outputs from different models removes any forms of interpretability. So the paper up SBM was great, and it showed very useful interpretability forms, but InterpGN makes the model a black box again...

- There are also some justified choices, for example:

    -  Why have $\mathcal{L}_{\text{div}}$ while ablation studies show it didn't really help in any way?
     -  Why use $\eta$ as a gating function and not a linear model?

- The code was not provided to replicate the experiments.

## Clarity  -- Medium
- Please see the questions section

**Questions:**

- How are the shapelets learned? i.e., how do you get $s_{k}^{m,l}$? Are they randomly initialized for each channel and learned through back prob?

- Text is unclear in section 3; in the paper, lines 156-160, it is mentioned "Existing methods gain different levels of interpretability by inputting interpretable features (Zuo et al., 2023) into a simple model such as a linear layer (Ma et al., 2020; Qu et al., 2024)
or SVM (Li et al., 2021). However, such approaches usually fail to provide explanations of their  predictions based on distance features. For the interpretable expert, we build logical predicates using shapelets, and the classifier directly provides rule-like explanations." From this paragraph, one can assume that the classifier on top of the shapelets is not a linear classifier but something else. But in equation 3, it was mentioned that a linear classifier was used on top of the shapelets, so it is a bit confusing...

- In figure Figure 5, what do different colors correspond to? Are they different classes?

- Typos:
    -  line 98 and line 285 interpretablity
    -  line 437 intepretability
    -  line 533 interoperable

---

> ### Author Response · Authors · 2024-11-21
> **Author Responses**
>
> Thank you very much for your valuable feedback and for taking the time to carefully read our paper. We address your questions below. If you would like further clarifications, we would be happy to continue discussions.
>
> > Regarding Weakness 1: interpretability of InterpGN and relationship with IME
>
> We appreciate the reviewer’s concerns regarding the interpretability of InterpGN compared to models like IME. We acknowledge that combining outputs from both SBM and DNN can introduce a level of opacity. However, we argue that the interpretability is not entirely lost. The gating mechanism, quantified by $\eta$, reflects how much the model relies on the interpretable SBM versus the DNN. As shown in Figure 7, when $\eta$ is high, the model’s predictions align closely with the SBM, preserving interpretability.
>
> We also appreciate the insightful comparison with IME. We fully agree that assigning a sample to a single expert during inference, as done in IME, can indeed maximize interpretability. To address this, we introduce a gating value $\underline{\eta}$ during inference of InterpGN, where only the SBM component is activated if $\eta(x_i)$ is above the gating value, i.e., $\eta(x_i) = 1$ if $\eta(x_i) > \underline{\eta}$ (see Section 4.5). With the exact training procedure as before, we experiment with $\underline{\eta} \in \{0.5, 0.6, 0.7, 0.8, 0.9, 1.0\}$. We observe that, although the average accuracy decreases if using a small $\underline{\eta}$, the difference between $\underline{\eta} = 0.5$ and $\underline{\eta} = 1$ is only 0.004 (see Section 5.3 and Figure 9). These results verify that our proposed approach can achieve a balance between interpretability and predictive accuracy, while enforcing interpretability maintains the same level performance.
>
> > Regarding model design choices
>
> - Diversity Loss: While the ablation study in Section 5.3 shows that  $\mathcal{L}_{\text{div}}$ does not have a significant impact on the main metrics, it encourages the model to learn less redundant shapelets. We think that it still introduces favorable behavior, and therefore, keep it in our model design.
>
> - Choice of Gating Function: One of the key contributions of our approach is using the SBM as both the interpretable expert and the gating mechanism. Instead of employing an additional linear assignment model as seen in IME, we base the gating on the confidence level ($\eta$) derived directly from the SBM. This design is more interpretable because the gating value directly reflects the model’s confidence in its interpretable predictions. Unlike a linear function, which may be simple but lacks semantic meaning, our approach ensures that the decision to rely on the DNN is driven by the SBM’s confidence. Additionally, our design also saves the $O(MT)$ parameters compared to having an extra linear gating function.
>
> > Regarding open-source implementations
>
> Thank you for pointing this out. We will release the full implementation and experiment scripts to ensure reproducibility of all the results presented in the paper.
>
> > We answer your questions below:
>
> 1. The shapelets are indeed model parameters that are learned through back-propagation. They are randomly initialized for each channel and optimized during training to capture discriminative patterns in the time-series data. We clarify this in the revised manuscript.
> 2. We believe the confusion stems from differentiating between building a linear classifier over distances (existing methods) vs predicates (as in SBM). We clarify this in the revised manuscript.
> 3. The different colors in Figure 5 correspond to different classes in the dataset. We add this clarification in the figure caption for better understanding.
> 4. Minor corrections: we thank the reviewer for pointing out these typos. We correct them accordingly in the revised manuscript.
>
> We thank the reviewer again for their feedback and hope that we have addressed their questions. If so, we hope they will consider increasing their score.

---

> > ### Comment · Reviewer_Ausa · 2024-11-25
> > **Than you for the rebuttal**
> >
> > Thank you for your response and for making so many adjustments in a short time period.
> > - The choice of gating function makes sense after your clarification.
> > - Having a confidant threshold at inference time does address *some* of my interpretability concerns; I am curious about the percentage of samples that are assigned to SBM vs DNN at different thresholds; if you can clarify that in the manuscript, that would be very helpful.
> >
> > Given the clarifications and inference thresholding experiments, I have raised my score.

---

> > > ### Author Response · Authors · 2024-11-27
> > > **Thank you**
> > >
> > > Thank you very much for taking our rebuttal into consideration. We have updated Figure 19 to include the percentage of samples assign to the SBM (i.e., $\eta(x_i) = 1$) under different gating values, as requested. We appreciate your constructive feedback and detailed comments.

---

### Official Review · Reviewer_Jqxe · 2024-11-06

**Soundness:** 3
**Presentation:** 2
**Contribution:** 2
**Rating:** 5
**Confidence:** 4

**Summary:**

This article proposes Shapelet Bottleneck Model  (SBM) a framework for time-series classification that produces an interpretable model. This framework is based on the adaptation of an existing model (Learning Time-Series Shapelets, LTS). The main idea is to modify shapelet learning by introducing a Gaussian kernel to measure the distance between the time-series and the shapelet, as well as a cost function to ensure diversity among the learned shapelets and greater sparsity in the model's use of shapelets. The learned shapelets are then provided to a linear classifier. An analysis of the weights allows identifying the shapelets that are important for classification. The second contribution is the introduction of a Mixture-of-Experts model and a gating function to ensure good classification performance when the first model is insufficient. The approach is evaluated on 30 common datasets compared to state-of-the-art algorithms.

**Strengths:**

* The paper addresses an important topic regarding the explainability of classification in time series.
* The evaluations conducted are numerous and compared to a sufficient number of state-of-the-art algorithms.
* The experiments are well analyzed, and the qualitative analyses are welcome.

**Weaknesses:**

* The state of the art is relatively brief, while there is a wealth of literature on the subject. However, the main works are well cited.
* The model section of the paper is relatively difficult to read. For SBM, even though it is an iteration of LTS, it would have been useful to say a few words about end-to-end gradient shapelet learning (at least a reference to the LTS paper in this section).
* The Mixture-of-Experts (MoE) section is really hard to follow due to the very limited space given to it. Very few details are provided about this part.
* The main contribution of the paper, given the stated objective — an interpretable time-series classification model — is the proposal of a Gaussian kernel distance rather than an Euclidean distance in the LTS approach, which seems rather modest.
* On the experimental side, given the similarity of the approach with LTS, why not use LTS as a baseline as well?
* Some results are unclear: what exactly do "Wins/Draws" and "Losses" mean in Table 1? What does the p-value refer to?
* Unless I am mistaken, there is no comparison with other interpretable methods for the interpretability results.

**Questions:**

Please see weaknesses.

---

> ### Author Response · Authors · 2024-11-23
> **Author Responses (part 1)**
>
> Thank you for your comments and feedback. We address your concerns below and provide the necessary clarifications. If you would like further clarifications, we would be happy to continue discussions.
>
> > Regarding the baseline methods
>
> Thank you for raising this point. In the revised manuscript, we have updated the list of baseline methods to align with the current state-of-the-art in deep learning methods. The performance of the new baselines is summarized in Table 1, with detailed results included in Table 6. If there are additional specific baselines you believe should be included, we would appreciate your suggestions.
>
>
> > Clarification on the training of SBM
>
> We have added a clarification in Section 4.2: "The parameters to be trained include the linear classifier weights $w_{c, k}^{m, l}$ and the shapelets $s_{k}^{m ,l}$, both of which are randomly initialized."
>
> > Clarification on the design of InterpGN
>
> To address potential confusion regarding the Mixture-of-Experts gating design in InterpGN, we provide a step-by-step clarification:
> 1. Given a time-series data $x_i$, the output (logit) of the SBM (interpretable expert) is $r_i$ and the output of the DNN expert is $z_i$.
> 2. Since the SBM is a classifier trained with a cross-entropy objective, $\hat{r}_i = \texttt{softmax}(r_i)$ is optimized toward a one-hot vector. Therefore, the sparsity of $\hat{r}_i$ can measure the confidence of SBM's predictions.
> 3. InterpGN is designed to use the SBM for samples with significant shapelet features while relying on the DNN to assist with challenging samples. Based on "the use of DNN should be inverse proportional to the confidence level of the interpretable model (Line 289)", we design the gating function $\eta(x_i)$ which measures the sparsity of $\hat{r}_i$.
> 4. In this setup, "the interpretable expert itself serves as the gating network" (Line 283), delegating work to the DNN based on its confidence. With gating value $\eta(x_i)$, the final output of the InterpGN is a mixture of $r_i$ and $z_i$, following conventional Mixture-of-Experts (MoE) methods.
>
> We hope this clarification resolves any confusion you may have. If you could provide details and sources of the confusion, we would be happy to clarify and discuss.
>
> > Regarding the contributions of this paper
>
> We would like to reclaim and clarify our contributions as outlined in the Introduction:
> 1. We introduce "a novel ating function that assigns samples to experts based on the confidence level of the interpretable expert", from which we build and train the InterpGN model. This is an essential contribution of this paper.
> 2. We propose "a variant of the Shapelet Transform" that improves interpretability over existing shapelet-based methods:
> - Unlike traditional Shapelet Transform methods that rely on shapelet distances, our formulation uses predicates measuring the likelihood of shapelet existence. This allows for rule-like predictions where classifier weights directly indicate the importance of the existence of specific shapelets for each class.
> - Our approach enables the automatic identification and ranking of essential shapelets, which is not feasible in distance-based methods.
> - This design also supports global and local explainability (Section 4.3).
> - As demonstrated in Section 5.3 (Table 9), the proposed RBF predicates improve both accuracy and shapelet quality compared to linear predicates.
> 3. We introduce quantitative metrics to measure interpretability and shapelet quality, whereas existing shapelet-based works primarily rely on selective visualization for qualitative demonstrations.

---

> ### Author Response · Authors · 2024-11-23
> **Author Responses (part 2)**
>
> > LTS in the experiments
>
> LTS is a foundational method for shapelet learning. Although it is not directly included as a baseline, we evaluate two advanced LTS variants, ShapeNet and ShapeConv, which use explicit shapelet initializations and regularization to improve upon the original LTS formulation. Their performance can be considered representative of LTS. In general, we would expect LTS to have similar performance as the SBM.
>
> In terms of interpretability, as noted in Sections 4.3 and 5.3, LTS methods based on shapelet distances share limitations with other approaches, as they cannot automatically identify essential shapelets based on classifier weights. In Section 5.3, we compare our formulation with linear predicates, which can be viewed as a variant of LTS, providing a fair and the most appropriate comparison to our method.
>
> > Regarding metrics in Table 1
>
> The metrics in Table 1 represent pairwise comparisons between InterpGN and a baseline method:
> - "Wins/Draws" counts datasets where InterpGN outperforms or performs equally with a baseline.
> - "Losses" counts datasets where InterpGN underperforms relative to the baseline.
> - The "p-value" is derived from the Wilcoxon signed-rank test, measuring statistical significance. A small p-value (close to 0) indicates significant performance differences, while a large p-value suggests comparable performance.
>
> These metrics are commonly used in classification literature, such as in SVP-T. Clarifications on these metrics are now included in Appendix A.1.
>
> > Comparison with other interpretable methods
>
> In terms of performance, we compare with several interpretable methods, including ShapeNet, RLPAM, ShapeConv, LTS with linear predicates, and SVP-T (weakly interpretable). However, as these methods employ different types of interpretable features, direct quantitative comparisons on interpretability are challenging. Evaluating qualitative results would require user studies, which may be subjective and is a major effort that we cannot complete at this time.
>
> Nevertheless, in Section 5.3, we compare with LTS using linear predicates, which serves as the most appropriate direct comparison to our method. Using quantitative metrics, we demonstrate that our approach outperforms this baseline.
>
> We appreciate your comments and hope these clarifications address your concerns effectively. If there are additional points you would like us to elaborate on, we welcome further discussion.

---

### Official Review · Reviewer_Vz3P · 2024-11-07

**Soundness:** 3
**Presentation:** 4
**Contribution:** 3
**Rating:** 6
**Confidence:** 3

**Summary:**

This paper introduces the Interpretability Gated Network (InterpGN), a time-series classification framework that implements interpretability with sufficient performance guarantees. InterpGN integrates a deep neural network and a concept-based shapelet bottleneck model (SBM) to model patterns for classification using interpretable shapelets (i.e., unique time-series subsequences). Here, linear classifiers are simple to interpret because the importance of each feature is directly reflected in the magnitude of its weight. By integrating predicates into a mixture of experts (MoE) framework, InterpGN uses a gating function to assign tasks based on the model's confidence.

**Strengths:**

1. The gated mechanism is novel. Such a gating function based on SBM's confidence level preserves interpretability for simple cases while calling on the DNN for complex samples.

2. The motivation for adding interpretability to time-series classification is compelling, specifically since the link between capturing temporal patterns and classification outcomes is often not intuitive.

3. New quantitative metrics are used to assess shapelet quality and interoperability, demonstrating improvements over existing models.

4. InterpGN achieves comparable or superior accuracy on benchmark datasets.

5. The manuscript is well-written, with a clear presentation of concepts and methodologies. Additionally, the authors effectively visualize the schematic concepts of the paper (Fig. 1, 2, and 4). These illustrations clearly present the core ideas.

**Weaknesses:**

1. Though interpretability is a primary focus, InterpGN reintroduces black-box elements due to the integration of DNN outputs. This can override shapelet-based explanations in challenging cases. The interpretability is particularly compromised when SBM fails to capture patterns adequately, and the model defaults to the DNN expert without sufficient transparency regarding how the shapelet information influences the final decision.

2. The authors claim that creating predicates using the RBF kernel allows for more flexible matching with the original time-series compared to the threshold distance-based shapelet approach. However, I'm not fully convinced that Fig. 3 effectively illustrates this intuition. In MTS classification, there could be relatively less important features (i.e., those that do not impact the classification results), so what's the justification for requiring a shapelet for every variable?

3. Interpretation of Fig. 6 may seem forced; the claim that the "OS" variable is steady appears subjective, as the left side actually seems steadier than the right side.

4. I'm entirely persuaded of the usefulness of "global explanation." As the authors explain well, local explanation is valuable as a post-hoc method for identifying the shapelets that most influenced the classification outcome. However, compared to recent time-series classifications that achieve high performance, what is the necessity of a global explanation? Further intuition on this point would be helpful.

5. The performance improvement in terms of average accuracy in Tab. 1 seems too small. Since InterpGN uses both FCN and SBM, i.e., more parameters, we expect the performance improvement to be significant. However, the improvement is only 0.6% in terms of average accuracy.

**Questions:**

1. How do we get shapelets? Is it a learnable parameter?

2. Why is the length of the shapelet considered as in the paper? Why is the minimum length fixed at 3?

3. Can InterpGN be applied to other time-series applications? I'm really curious to see if InterpGN performs well in time-series anomaly detection (since it's a one-class classification) or some future prediction tasks.

4. Where and how were the hyperparameters introduced in Eq. (5) chosen?

5. Fig. 5 (c) is interesting, but I have a question: low $\eta$ (low transparency; low confidence of expert) data relies on the DNN output. However, line 388 states, 'Intuitively, predictions of opaque points are interpretable as they are based on SBM, while transparent points rely more on the DNN,' which is confusing.

6. Considering that the MIMIC-III in-hospital mortality dataset is imbalanced, how does InterpGN address potential issues in capturing representative shapelets for the minority class, and does it risk overfitting the majority class?

7. Why was an ablation study on the effect of L (length candidates of predicates)?

8. As the authors highlighted in the limitations section, a drawback of InterpGN is that its interpretability is somewhat rigid (e.g., sequences belonging to a certain class must have or must not have certain predicates). Given this, could InterpGN be utilized as a feature selection tool to enhance classification performance rather than the interpretability tool?

---

> ### Author Response · Authors · 2024-11-21
> **Response to the weaknesses**
>
> Thank you very much for your valuable feedback and for taking the time to carefully read our paper. We address your questions below. If you would like further clarifications, we would be happy to continue discussions.
>
> > Regarding Weakness 1
>
> The integration of a DNN in InterpGN is motivated by the limitations of predefined interpretable features like shapelets, which may lack the expressiveness to predict marginal and complex samples (Line 82). The DNN is introduced to enhance the predictions for cases where shapelet features alone are insufficient. For example, in Figure 5 (c), shapelet features are sufficient to predict the shaded samples, while the lighter samples near the decision boundary may need extra help from the DNN.
>
> The design of InterpGN maintains interpretability for samples with significant shape-level feature, while selectively engaging the DNN when needed. Therefore, the interpretability of InterpGN is sample-specific, preserving transparency when possible while improving performance on difficult cases. Note that for challenging cases with low $\eta$, one would not worry that the InterpGN defaults to the DNN since the interpretable features from the SBM may not be useful in the first place (i.e., we would not trust predictions from the SBM with low condidence).
>
> Finally, we would like to argue that the combination of an interpretable model and a DNN is not a weakness but a strength to compensate for the limitations of interpretable models. The mixture-of-expert design in InterpGN enhances the predictive performance while preserving interpretability when possible.
>
> > Regarding Weakness 2
>
> Figure 3 is intended to provide a qualitative comparison, which aligns with standard practices in shapelet-based literature. Beyond this, we have also introduced a quantitative metric to assess shapelet quality (Equation 9) and validated the advantages of using RBF-based predicates with quantitative results in Table 9.
>
> Regarding the inclusion of shapelets for every variable, it is important to note that these shapelets are model parameters learned from scratch. Initially, shapelets are defined for all variables as an initialization strategy, ensuring broad coverage. However, through training, the model uses L1 regularization to enforce sparsity, allowing the linear classifier to automatically select only the most informative features. As shown in Section 5.3, the final learned model is often sparse.
>
> > Regarding Weakness 3
>
> We appreciate the reviewer’s observation regarding the interpretation of Figure 6. We acknowledge that the explanations provided may appear subjective, particularly given the lack of domain-specific expertise. The purpose of Figure 6 is to showcase the interpretability of our model using shapelet-based explanations. We agree that "steady" is the wrong wording for the "OS" variable but rather that the patient didn't survive; OS has longer trends (down and up) while the patient that did survive has short fluctuations around the same mean. In practice, these interpretations would be further refined and validated by domain experts to ensure alignment with real-world understanding.
>
>
> > Regarding Weakness 4
>
> Thank you for raising this point. The value of global explanations lies in providing a sample-independent understanding of the model’s learned decision rules. Unlike local explanations, which only explains the reason behind individual predictions, global explanations reveals the overall patterns and features that the model is looking for across the dataset. In practice, a global explanation can highlight generalizable insights about the dataset. Specifically, if we consider Figure 4, as suggested in the caption, the classifier can be interpreted as a "CW Circle" gesture should contain the blue shapelet but not the red shapelet in channel 1. This is an insight that applies to all instance of class "CW Circle", but of course, it can be violated in local explanations, for example, by challenging border instances.
>
> > Regarding Weakness 5
>
> Thank you for your feedback. We acknowledge that the performance improvement in terms of average accuracy is modest, but we argue this should not be surprising. Since SBM is a linear model over shapelets (which are learned), SBM has less freedom than FCN so we are also not surprised that it underperforms FCN. Furthermore, it only underperforms by 2.2\% which we find more surprising and, moreover, encouraging. The hope in adding interpretability with SBM is to at least maintain the performance of the most specialized and uninterpretable model, which is demonstrated by Table 1. A slight improvement in performance is likely due to overfitting of FCN on instances where the simpler SBM is sufficient.

---

> ### Author Response · Authors · 2024-11-21
> **Response to the questions**
>
> > We answer your questions below:
>
> 1. Yes, shapelets are trainable model parameters. They are randomly initialized and optimized through gradient descent to capture discriminative features within the data. Specifically, $s_k^{m, l} \in \mathbb{R}^{l}$ are the trainable parameters that appear in equation (2) defining the Shapelet Transform. We add a clarification in Section 4.2.
> 2. Thank your for this question. In principle, our proposed models could be adapted to tasks where shapelets are useful. For example, [1] presented a method of using shapelets on anomaly detection, while the use of shapelets in forecasting seems unexplored. However, as you suggested below, if we use the models as feature selection tools, it is possible to extend them to forecasting tasks by predicting the future shapelets. We envision these extensions as valuable future works.
> 3. As described in Section 3, we use shapelets of varying lengths (ranging from 5\% to 80\% of the time-series length) to capture features at different scales. The minimum length is set to 3 timestamps because shorter sequences would either represent a single point (1 timestamp) or a straight line (2 timestamps), which can be too simple to be considered as shapes.
> 4. The default hyperparameters are described in Appendix A.1. Rather than fine-tuning them for each dataset, we focused on studying their impact on accuracy, shapelet quality, and interpretability in Section 5.3.
> 5. To clarify: the transparency level in Figure 5(c) reflects the model’s confidence. Opaque points indicate high confidence (high $\eta(x_i)$) in the SBM’s interpretable predictions, while more transparent points indicate lower confidence (low $\eta(x_i)$), leading to a greater reliance on the DNN expert for those samples. (The confusion might be that the transparency of the scatter plot is inversely correlated with transparency of the model). In the revised manuscript, to address this confusion, we update the caption of Figure 5, and replace "opaque" and "transparent" with "shaded" and "lighter".
> 6. As detailed in Appendix A.2, we use a balanced subset of the MIMIC-III dataset to mitigate class imbalance. While we acknowledge that overfitting to the majority class can still be an issue, our primary focus was on demonstrating interpretability. Existing approaches, such as explicit class-balancing loss, can be integrated with our models to address this issue.
> 7. The range of shapelet lengths (5\% to 80\% of the series length) is intended to provide an initial universal setup for diverse datasets. Different datasets may have essential features at varying scales, so this setup serves as a general starting point. Therefore, we focuses on other more essential hyperparameters in the ablation study.
> 8. Thank you for this suggestion. We agree that InterpGN could be used as a feature selection tool. Typically, for such applications, one would want to retrain the model on the selected features, which in this case are the selected shapelets. Of course, we must also acknowledge that enhanced classification performance would be subject to the types of features, and in our experiments, we observe only a small increase in performance as you noted above.
>
> [1] Beggel et al., "Time series anomaly detection based on shapelet learning", Computational Statistics, 2019.
>
> We thank the reviewer again for their feedback and hope that we have addressed their questions. If so, we hope they will consider increasing their score.

---

> > ### Comment · Reviewer_Vz3P · 2024-11-22
> > **Thank you for your response**
> >
> > Thank you for your response. I really appreciate your addressing my concerns and revising the manuscript.
> >
> > - First, my main concern was that InterpGN might reintroduce DNNs, which might hinder interoperability. However, the authors have clearly addressed my concerns. I would like to suggest that they distinguish between simple cases that rely heavily on shape-level features and harder cases that use more DNNs, which would be helpful for readers and the community.
> >
> > - I would also like to see (if possible) examples of extending InterpGN to other applications, such as time series prediction or anomaly detection, where clear quantitative results can be obtained. This may be out of scope, but I think it would make the authors' work stand out even more.
> >
> > In this respect, I think this paper can contribute to the ICLR community, and I have raised the acceptance score. Once again, I would like to thank the authors for their hard work.

---

> ### Author Response · Authors · 2024-11-25
> **Thank you!**
>
> We thank you again for taking our responses into consideration. We greatly appreciate your time and efforts to review our work and provide constructive feedback.

---

### Meta-Review · Area_Chair_8rsg · 2024-12-17

**Metareview:**

This paper introduces a novel Mixture of Experts framework that combines a deep neural network (DNN) with an interpretable Shapelet Bottleneck Model for time-series prediction. This approach offers "partial" interpretability by combining the outputs of both models to generate the final prediction.  The authors demonstrate compelling performance on the MIMIC-III dataset while providing interpretations.

The proposed method successfully balances predictive accuracy with interpretability, achieving comparable performance to state-of-the-art time-series prediction models. By combining existing techniques in a novel way, the authors address a critical need in the field. Importantly, they clearly acknowledge the limitations of their approach, helping readers understand the specific scenarios where this method can be most beneficial.

However, as other reviewers have pointed out, the combination of DNN and interpretable models for the final prediction inevitably leads to a loss of some interpretability. This trade-off should be explicitly addressed in the camera-ready version. Additionally, the performance improvements appear marginal, and there are concerns about the scalability of the proposed method.

Despite these limitations, the paper offers significant value to the community by providing a practical approach for achieving partial interpretability in time-series prediction while maintaining competitive performance. Therefore, I recommend accepting this paper.

**Additional Comments On Reviewer Discussion:**

The authors effectively addressed the reviewers' concerns during the rebuttal period. They acknowledged certain limitations and clarified them in the revised manuscript, demonstrating a willingness to engage with the feedback.  Their clarification regarding "partial" interpretability helped alleviate concerns in this area.

To further enhance the paper's impact, I recommend including additional empirical results beyond the MIMIC-III dataset. This would provide a more comprehensive understanding of the method's capabilities and generalizability.

---

### Decision · Program_Chairs · 2025-01-22

Accept (Poster)